# MULTIMODAL UNSUPERVISED DOMAIN GENERALIZATION BY RETRIEVING ACROSS THE MODALITY GAP

**Christopher Liao**
Boston University
cliao25@bu.edu

**Christian So**
Boston University
cbso@bu.edu

**Theodoros Tsiligkaridis**
MIT Lincoln Laboratory
ttsili@ll.mit.edu

**Brian Kulis**
Boston University
bkulis@bu.edu

## ABSTRACT

Domain generalization (DG) is an important problem that involves learning a model which generalizes to unseen test domains by leveraging one or more source domains, under the assumption of shared label spaces. However, most DG methods assume access to abundant source data in the target label space, a requirement that proves overly stringent for numerous real-world applications, where acquiring the same label space as the target task is prohibitively expensive. For this setting, we tackle the multimodal version of the *unsupervised domain generalization* (MUDG) problem, which uses a large *task-agnostic unlabeled* source dataset during fine-tuning. Our framework relies only on the premise that the source dataset can be accurately and efficiently searched in a joint vision-language space. We make three contributions in the MUDG setting. Firstly, we show theoretically that cross-modal approximate nearest neighbor search suffers from low recall due to the large distance between text queries and the image centroids used for coarse quantization. Accordingly, we propose *paired k-means*, a simple clustering algorithm that improves nearest neighbor recall by storing centroids in query space instead of image space. Secondly, we propose an adaptive text augmentation scheme for target labels designed to improve zero-shot accuracy and diversify retrieved image data. Lastly, we present two simple but effective components to further improve downstream target accuracy. We compare against state-of-the-art name-only transfer, source-free DG and zero-shot (ZS) methods on their respective benchmarks and show consistent improvement in accuracy on 20 diverse datasets. Code is available: https://github.com/Chris210634/mudg

## 1 INTRODUCTION

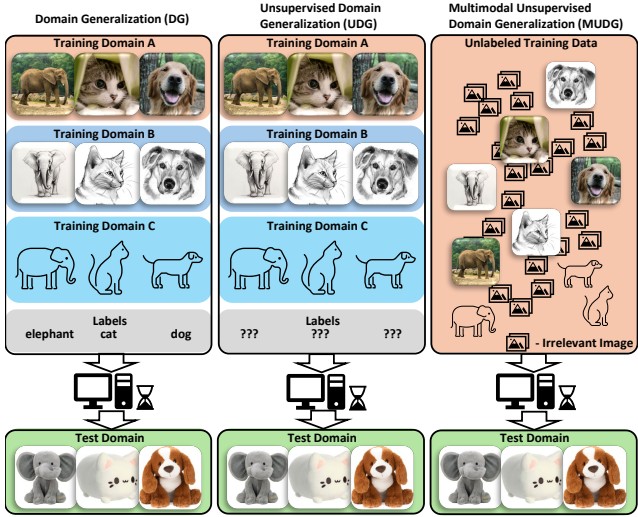

Figure 1: Comparison of Problem Settings. DG assumes labeled source data for the target task. UDG assumes the same data, but without labels. Our MUDG setting assumes a superset of the unlabeled source data, most of which is irrelevant to the target task. The core challenge of MUDG is constructing a task-specific dataset for model finetuning.

*Domain generalization (DG)* is widely studied in the computer vision literature because the train and test image data distributions often differ for many applications. However, traditional DG methods assume access to labeled task-specific source data, which is expensive for many real-world applications. Consequently, more recent studies have tackled the unsupervised DG (UDG) problem, where source labels are not used during finetuning (Zhang et al., 2022; Narayanan et al., 2022; Harary et al., 2022). Unfortunately, this experimental procedure is fairly restrictive and impractical, since it still assumes that the source and target label spaces are identical. To address this shortcoming, we propose studying a more realistic multimodal UDG (MUDG) setting, where the source data is both unlabeled and "task-agnostic", i.e. the source dataset is not specifically designed for the

target task. Instead, we only require that the source dataset contains the target visual concepts, and that these target visual concepts are aligned with the corresponding target language concepts in the

| Setting | Task-agnostic Source | Task-specific Source | Unlabeled Target | Target Label Names |
|---|---|---|---|---|
| DA (Sun et al., 2016; Saito et al., 2019; Acuna et al., 2021) | - | labeled | ✓ | ✓ |
| SFDA | - | - | ✓ | ✓ |
| ZS (Menon and Vondrick, 2022; Pratt et al., 2023; Roth et al., 2023; Novack et al., 2023) | - | - | - | - |
| DG (Cha et al., 2022; Min et al., 2022; Shu et al., 2023; Khattak et al., 2023) | - | labeled | - | ✓ |
| UDG (Zhang et al., 2022; Narayanan et al., 2022; Harary et al., 2022) | - | unlabeled | - | ✓ |
| SFDG (Cho et al., 2023) | - | - | - | ✓ |
| **MUDG** (Udandarao et al., 2023) (our work) | ✓ | - | - | ✓ |

Table 1: Comparison to related works based on the information available at training time. DA - domain adaptation; DG - domain generalization; SF - source free; ZS - zero-shot. UDG - unsupervised DG. MUDG - multimodal UDG is our setting.

CLIP embedding space used for indexing. Under these relaxed assumptions, we can leverage publicly available large-scale image datasets to improve DG accuracy by building a subset of images relevant to the target task. Figure 1 compares our proposed MUDG with the most relevant problem settings.

**Multimodal Unsupervised Domain Generalization (MUDG)** In order to leverage publicly available unlabeled image data, such as LAION (Schuhmann et al., 2022), YFCC100M Thomee et al. (2016), WIT Srinivasan et al. (2021), and CC12M Changpinyo et al. (2021), we study MUDG, a generalization of UDG classification. "Multimodal" refers to the requirement for the source dataset to be accurately and efficiently searchable in a joint vision-language space using a pretrained CLIP model. Using this searchable index, our goal is to build a pseudo-labeled subset of the source data to train a model for a given target classification task. Table 1 positions our problem setting relative to related works. Our work can be viewed as an extension of the recent source-free domain generalization (SFDG) problem (Cho et al., 2023), which only uses the target label names during finetuning. Compared to SFDG, finding a suitable subset of the source data poses an interesting challenge, and our results show that the margin for accuracy improvement is much larger under our MUDG setting. MUDG is most similar to the "name-only transfer" problem posed by SuS-X Udandarao et al. (2023), but our work is more in line with the DG literature.

To illustrate the practical benefits of MUDG over UDG, consider a set of e-commerce classification tasks (e.g. "casual" vs. "formal", "modern" vs. "traditional"). In the traditional UDG setting, a dataset must be curated for each category in each classification task. In contrast, MUDG allows for the use of *one* universal unlabeled dataset of general product images. We accomplish this by selecting a relevant and representative subset of the general dataset using the target class names, and then training a customized model for each task. This approach is more practical and scalable, since it eliminates the need for handcrafted task-specific datasets.

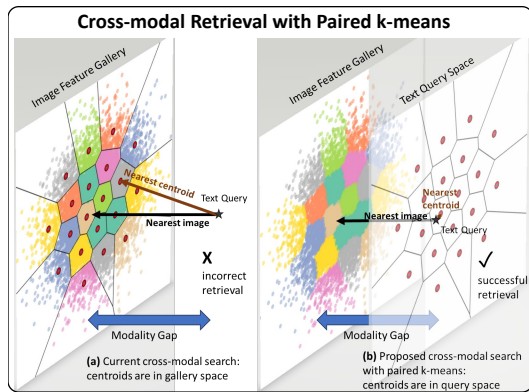

Figure 2: Illustration of Paired k-means. In this example, image samples are grouped using k-means, and the search is limited to the group whose centroid is closest to the query. Left: Due to the large distance between the query and image samples, the true nearest neighbor is unlikely to reside within the Voronoi cell of the closest image centroid. Right: We correct this issue by maintaining the centroids of the Voronoi cells in query space.

**Accurate and Efficient Retrieval** The core challenge of MUDG is accurate approximate nearest neighbor search for finding images relevant to the target task. Similar to retrieval augmentation (Blattmann et al., 2022; Gur et al., 2021; Long et al., 2022; Iscen et al., 2023), we propose constructing a subset of images retrieved from the source dataset using the text query "a photo of a ⟨label⟩". Existing works use an off-the-shelf inverted feature list (IVF), which organizes images into buckets based on the closest feature centroid, as shown in Figure 2. During deployment, the index calculates similarity scores between the query and images only in the bucket corresponding to the closest centroid. This simple search algorithm works well when the probability of the query and its nearest neighbor residing in the same Voronoi cell is high. However, we show theoretically that this probability is low when the query belongs to a different modality due to the well-known modality gap (Liang et al., 2022; Oh et al., 2024; Shi et al., 2023; Ming and Li, 2024). We empirically

confirm that cross-modal approximate nearest neighbor search using an IVF index has lower recall than in-modal search. In other words, a query may return images that belong to a different label, leading to low downstream target accuracy. To mitigate this issue, we propose *paired k-means*, a clustering algorithm that maximizes the probability of a text query and its closest image sample belonging to the same Voronoi cell by updating the centroids to be in the query distribution. We show empirically that paired k-means converges and leads to better cross-modal recall under the same latency constraint.

**Diversified Retrieval**  On the other hand, accurate retrieval is not sufficient for high target accuracy, since a training dataset that covers only the small, high-confidence region of the target image distribution is undesirable. To introduce diversity, we must augment the text query, e.g. "a photo of a chicken, ⟨descriptor⟩." Pratt et al. (2023) and Menon and Vondrick (2022) use LLM-generated descriptors to augment the query, but Roth et al. (2023) suggest that these LLM descriptors achieve the same zero-shot accuracy as random text augmentations. Intuitively, querying with descriptors that already achieve high zero-shot accuracy should lead to better target performance after finetuning. Following this intuition, we design an unsupervised heuristic to select good label augmenta-

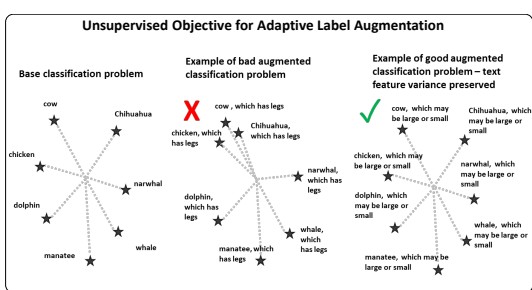

Figure 3: Adaptive Label Augmentation. Starting with a base classification problem, our goal is to find label augmentations that preserve the variance of the classifier prototypes. Intuitively, maintaining feature variance enhances zero-shot accuracy by better capturing the underlying data distribution.

tions adaptively based on the target classification task, without an LLM or image data. Our heuristic favors augmentations that do not reduce the variance between target text features (see Figure 3). We show that our adaptive descriptor selection achieves state-of-the-art zero-shot accuracy across 10 standard datasets, and that this translates to additional gains in downstream target accuracy.

Finally, we introduce two additional components that makes our method more robust to irregularities in the source data and the target task: (1) Sample selection by clustering: cluster image embeddings into $k$ clusters within each label group and randomly select one sample from each cluster. The purpose of this step is twofold: to build a balanced dataset, with $k$ samples per label; and to ensure that no two images are semantically similar. (2) Diversity preserving loss: regularize the KL divergence between current and initial soft predictions on training samples for every augmentation to avoid collapse of textual representations.

Overall, we make two core contributions related to retrieval in the context of MUDG:

- We identify a fundamental limitation of cross-modal approximate nearest neighbor search caused by the modality gap. We investigate this challenge theoretically, and propose a paired k-means clustering algorithm for building an index with better cross-modal recall.
- We develop an unsupervised adaptive label augmentation scheme for diversified retrieval.

## 2  RELATED WORK

**Multimodal foundational models**  Multimodal foundational models (Radford et al., 2021; Jia et al., 2021; Li et al., 2022b; Yu et al., 2022) use separate image and language encoders to embed the two modalities into a joint space. Once pretrained, these embeddings can be used to create a database searchable by both image and text (Schuhmann et al., 2022). Large-scale efficient search is enabled by approximate nearest neighbor search libraries such as FAISS (Douze et al., 2024). A recent work (Gadre et al., 2023) achieved the latest state-of-the-art on ZS ImageNet by cleaning LAION-5B with a teacher CLIP model. Another recent work (Sun et al., 2023) uses a large CLIP model and unpaired web-crawled data to train a smaller foundational model in a distillation-inspired manner. The above works focus on generalist pretraining from scratch, which remains out-of-reach of most academic researchers. We focus instead on task-specific finetuning using a constructed dataset of up to 100K samples. React (Liu et al., 2023) tackles the so-called "model customization" problem; in comparison, our work is more focused on the source subset construction portion of the finetuning pipeline, and consequently, we achieve similar accuracy improvements as React with a 100× smaller retrieved dataset. Our problem setting is most similar to SuS-X (Udandarao et al., 2023), which retrieves a support set from LAION-5B, but they focus on the training-free regime. Many recent

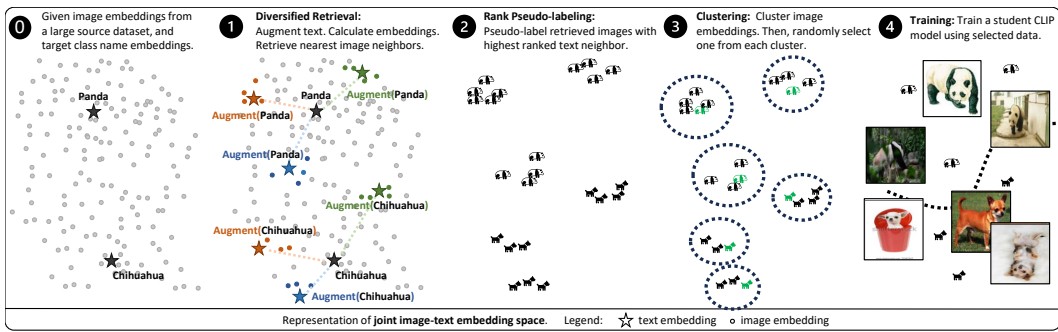

Figure 4: Illustration of Our Training Pipeline. In step 1, we select good label augmentations according to Fig. 3 and retrieve their nearest neighbors using the procedure shown in Fig. 2. In step 2, we pseudo-label the retrieved images with the closest label feature. In step 3, we de-duplicate the retrieved dataset by clustering and then selecting one image from each cluster. Finally, we finetune the CLIP classifier on the dataset.

works strive to understand and tackle the modality gap (Liang et al., 2022; Oh et al., 2024; Shi et al., 2023; Ming and Li, 2024) in the context of model transfer; unlike these works, we study the modality gap's implications on cross-modal search. Ming and Li (2024); Iscen et al. (2023); Liu et al. (2023) work around the cross-modal retrieval problem by additionally performing in-modal search, which is not possible for every application; we work towards more effective cross-modal search.

**Flavors of domain generalization**    Table 1 is a non-exhaustive summary of variations on generalization settings studied in recent literature. Domain adaptation (Sun et al., 2016; Saito et al., 2019; Acuna et al., 2021) aims to leverage out-of-distribution (OOD) but task-specific source data in conjunction with unlabeled target data. Traditional DG (Muandet et al., 2013; Cha et al., 2022; Min et al., 2022) trains on OOD task-specific source data from multiple domains, without knowledge of target data. A more recent flavor of DG (Shu et al., 2023; Khattak et al., 2023) trains on generic labeled source data (e.g. ImageNet) with the goal of generalizing to any classification task, by leveraging transferability of the image-text alignment in CLIP. Unsupervised DG (Zhang et al., 2022; Narayanan et al., 2022; Harary et al., 2022) trains on unlabeled task-specific source data. Source-free DG (SFDG) (Cho et al., 2023) aims to increase pretrained accuracy with only the target task information, but the improvement over ZS methods is not consistent empirically. ZS methods (Menon and Vondrick, 2022; Pratt et al., 2023; Roth et al., 2023; Novack et al., 2023) improve accuracy by ensembling multiple text features. Our problem setting, multimodal UDG, takes advantage of plentiful unlabeled non-task-specific image data, which offers more leverage than the SFDG setting, while not relying on any task-specific or labeled images contrary to the DA and DG studies.

**Webly supervised, open world, and open set**    The webly supervised literature (Chen and Gupta, 2015; Li et al., 2020) focuses on learning from a noisy web-crawled dataset (Li et al., 2017b; Sun et al., 2021) and is very closely related to the large body of work on noisy supervised learning, see survey (Song et al., 2022). These works focus on the finetuning algorithm given a dataset, rather than the construction of the training data, unlike our work. Another popular research direction focuses on generalization to unseen classes given a certain set of (possibly related) training classes; these works fall under open-set (Saito et al., 2021; Du et al., 2023; Panareda Busto and Gall, 2017) open-world (Bendale and Boult, 2015; Boult et al., 2019; Cao et al., 2022) or base-to-novel (Zhou et al., 2022; Khattak et al., 2023; Kan et al., 2023) semi-supervised learning. Finally, some works selectively retrieve from an unlabeled data pool to expand a smaller set of labeled training samples (Sener and Savarese, 2017; Killamsetty et al., 2021; Kim et al., 2023), referred to as core-set sampling. Contrary to these works, we assume no labeled data of any kind at training time. Retrieval augmentation (Blattmann et al., 2022; Gur et al., 2021; Long et al., 2022; Iscen et al., 2023) is a related line of work which requires a retrieval system at test time, adding substantially heavier evaluation overhead.

## 3    METHOD

Figure 4 illustrates our method. Concisely, we use augmented copies of the target label names to query a large source dataset and then finetune the student CLIP model on retrieved images, with the ultimate objective of high target accuracy. Toward achieving this objective, we identify two necessary sub-goals: building a search index with good cross-modal recall and designing a label augmentation scheme with high ZS accuracy. Section 3.1 tackles the first sub-goal with a novel cross-modal

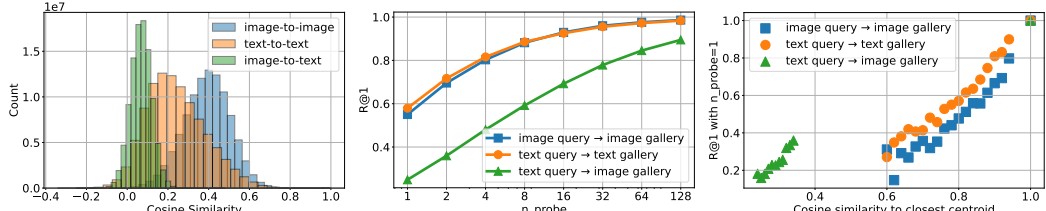

Figure 5: Left: Empirical confirmation of the modality gap; cross-modal similarity scores are lower than in-modal similarity scores. Middle: Cross-modal nearest neighbor search suffers from lower recall than in-modal search. Right: Empirical verification of Theorem 1; queries that are farther away from the closest centroid have lower recall.

indexing scheme for accurate and efficient retrieval; Section 3.2 solves the second sub-goal with an unsupervised heuristic to select good descriptors for augmenting label names; Section 3.3 finishes with a diversity preserving loss for model finetuning.

### 3.1 MORE ACCURATE CROSS-MODAL RETRIEVAL

**Background**    For this paper we will consider a two-level IVF indexing scheme used by Udandarao et al. (2023); Liu et al. (2023); Iscen et al. (2023). The first level is a coarse quantization consisting of $k$ buckets obtained by k-means clustering; the index stores the coordinates of the centroids and a list of sample IDs belonging to each bucket along with their residual features. The second level is a fine quantization scheme used to reduce disk storage. We will consider only the coarse quantization scheme. During deployment, the index sorts the centroids by decreasing similarity with the query and searches through the first $n_{\text{probe}}$ buckets for its nearest neighbor. Assuming that each bucket contains a similar number of samples, the query speed is proportional to $n_{\text{probe}}$. The quality of the index can be measured by the percentage of queries where the true nearest neighbor among all gallery samples is retrieved, "R@1", for constant $n_{\text{probe}}$.

**Motivational Issue**    Figure 5 Middle shows that the R@1 for text-to-image searches is about 30% lower than in-modal queries for small $n_{\text{probe}}$. This is a concern for many multimodal applications, since cross-modal retrieval exhibits a far worse recall-latency tradeoff than in-modal retrieval using existing technology. We hypothesize that this drop in recall is caused by the modality gap (Liang et al., 2022; Oh et al., 2024; Shi et al., 2023; Ming and Li, 2024), illustrated in Figure 5 Left. Text queries tend to be far away from the image centroids. Moreover, on a closed space, points far away from centroids are closer to the boundaries of the Voronoi cells, and the neighbors of boundary points are more likely to reside in neighboring cells.

**Assumptions 1** Consider $n$ points $\{\mathbf{x}_1, ..., \mathbf{x}_n\}$ drawn uniformly from the unit sphere $\mathcal{S}^d := \{\mathbf{x} \in \mathbb{R}^d \mid \|\mathbf{x}\|_2 = 1\}$. Consider $k$ additional points $\{\mathbf{c}_1, ..., \mathbf{c}_k\}$ drawn uniformly from $\mathcal{S}^d$. We refer to these points as "centroids"; $k << n$. The Voronoi cell around a centroid is the set of all points closer to that centroid than all other centroids, i.e. $\text{Vor}(\mathbf{c}) := \{\mathbf{x} \in \mathcal{S}^d \mid \|\mathbf{x} - \mathbf{c}\|_2 \leq \|\mathbf{x} - \mathbf{c}_i\|_2, \forall \mathbf{c}_i \sim \{\mathbf{c}_1, ..., \mathbf{c}_k\}\backslash\mathbf{c}\}$. We assume that $\text{Vor}(\mathbf{c})$ is a strict subset of the hemisphere centered at $\mathbf{c}$.

**Theorem 1 (Decreasing recall on closed space)** Under Assumptions 1, $\forall \mathbf{c} \in \{\mathbf{c}_1, ..., \mathbf{c}_k\}, \mathbf{p} \in \text{Vor}(\mathbf{c})$:

$$g_{\mathbf{c}}(\mathbf{p}) \leq \Pr\left[\underset{\mathbf{x}_i \sim \{\mathbf{x}_1, ..., \mathbf{x}_n\}}{\arg\min} \|\mathbf{x}_i - \mathbf{p}\|_2 \in \text{Vor}(\mathbf{c})\right] \leq g_{\mathbf{c}}(\mathbf{p}) + \epsilon \tag{1}$$

where $\epsilon := 1 - \rho(s')$; $\rho(\cos(\theta)) := \frac{1}{2}I_{\sin^2\theta}\left(\frac{d-1}{2}, \frac{1}{2}\right)$; $I_x(\cdot, \cdot)$ is the regularized incomplete beta function; and $g_{\mathbf{c}}(\mathbf{p})$ is a function defined over $\text{Vor}(\mathbf{c})$ which satisfies the following properties:

1. $g_{\mathbf{c}}(\mathbf{c}) = \rho(s')$, where, $s' := \cos\left(\frac{1}{2}\cos^{-1}\max_{\mathbf{c}_i \in \{\mathbf{c}_1, ..., \mathbf{c}_k\}\backslash\mathbf{c}}\langle\mathbf{c}, \mathbf{c}_i\rangle\right)$.
2. $g_{\mathbf{c}}(\mathbf{b}) + \epsilon = \frac{1}{2}$ for all points $\mathbf{b}$ on the boundary of set $\text{Vor}(\mathbf{c})$.
3. $g_{\mathbf{c}}$ is non-increasing in all directions from $\mathbf{c}$ in the following sense:

$$g_{\mathbf{c}}\left(\text{proj}_{\mathcal{S}^d}(a\mathbf{u} + \mathbf{c})\right) \leq g_{\mathbf{c}}\left(\text{proj}_{\mathcal{S}^d}(b\mathbf{u} + \mathbf{c})\right), \forall a > b > 0, \mathbf{u} \in \mathbb{R}^d$$

given that all inputs to function $g_{\mathbf{c}}$ remain within $\text{Vor}(\mathbf{c})$. $\text{proj}_{\mathcal{S}^d}$ denotes L2-normalization.

The proof follows from the convexity of Voronoi regions, see Appendix A.1. Note that $\rho(\cos(\theta))$ denotes the surface area of a spherical cap with angle $\theta$ as a fraction of the unit sphere's surface area (Li, 2010), and $\rho(s')$ is close to 1. In plain words, Theorem 1 states that under Assumptions 1, the

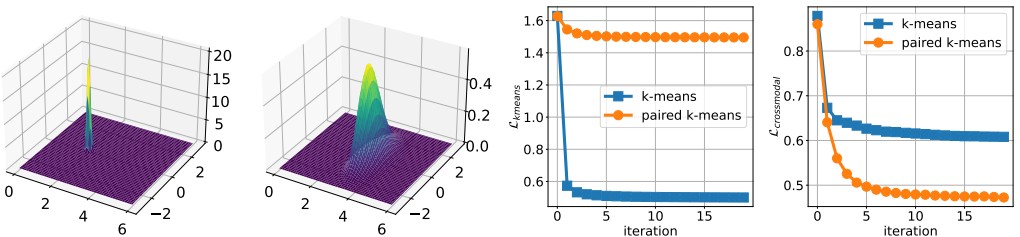

Figure 6: Left: Plots of the PDF in Eq. 2 of Theorem 2 for $d = 2$, illustrating both the small variance regime when $\mathbf{p}$ is in-distribution and the large variance regime when $\mathbf{p}$ is out-of-distribution. Right: Convergence of the two objectives in Eq. 5; note that the paired k-means algorithm is better at minimizing $\mathcal{L}_{\text{cross-modal}}$, the rate of cross-modal search failures on training data.

bounds on the probability that the nearest neighbor resides in the same Voronoi cell as the query decrease monotonically in all great circle directions from the centroid. This probability is equivalent to R@1 with $n_{\text{probe}} = 1$. The assumptions are somewhat stringent, but Figure 5 Right empirically verifies this behavior with a 20M subset of LAION-2B.

Theorem 1 partially explains the empirical observations in Figure 5 Middle, and text queries are certainly far away from their image centroids. However, Theorem 1 assumes that the query belongs to the same distribution as the gallery set. This is clearly not true for cross-modal retrieval. To understand the drop in recall when the query is not in the support of the gallery distribution, we need another set of assumptions. Theorem 2 will show that as a query moves away from a Gaussian distributed gallery distribution, the probability that the closest gallery sample and the closest centroid are close decreases. In fact, the distribution of the closest gallery sample approaches a Gaussian distribution in all except one dimension in the limit, i.e. if a query is far away, its position provides little information about the location of the closest gallery sample.

**Assumptions 2** Consider $n$ points drawn uniformly from $\mathcal{N}(\mathbf{0}, \mathbf{I}_d)$, the standard normal distribution in $\mathbb{R}^d$. Denote as $\{\mathbf{x}_1, ..., \mathbf{x}_n\}$. Let $\mathbf{q}(\mathbf{p}) := \arg\min_{\mathbf{x}_i \sim \{\mathbf{x}_1, ..., \mathbf{x}_n\}} \|\mathbf{x}_i - \mathbf{p}\|_2$.

**Theorem 2** Under Assumptions 2, the probability density function of the closest point to query $\mathbf{p}$ is:

$$\Pr[\mathbf{q}(\mathbf{p}) = \mathbf{x}] = n\left(1 - \Pr[\mathbf{x}_i \in \mathcal{B}_r(\mathbf{p})]\right)^{n-1}(2\pi)^{-d/2}\exp\left(-\frac{1}{2}\|\mathbf{x}\|_2^2\right), \; r := \|\mathbf{x} - \mathbf{p}\|_2 \quad (2)$$

where $\Pr[\mathbf{x}_i \in \mathcal{B}_r(\mathbf{p})]$ indicates the probability that a single point drawn from $\mathcal{N}(\mathbf{0}, \mathbf{I}_d)$ resides in the $\mathbb{R}^d$ ball of radius $r$ centered at $\mathbf{p}$.

**Corollary 2** The probability density function derived in Theorem 2 satisfies the following:

1. (Small variance when $\|\mathbf{p}\|_2$ is small)

$$\Pr[\|\mathbf{q}(\mathbf{p}) - \mathbf{p}\|_2 > r] \leq \left(1 - \left(\frac{r^d}{2^{d/2}\Gamma\left(\frac{d}{2} + 1\right)}\exp\left(-\frac{1}{2}(\|\mathbf{p}\|_2 + r)^2\right)\right)\right)^n \quad (3)$$

2. (Large variance when $\|\mathbf{p}\|_2$ is large).

$$\lim_{\|\mathbf{p}\|_2 \to \infty} \Pr[\mathbf{q}(\mathbf{p}) = \mathbf{x}] = n\left(\Phi\left(\|\text{proj}_{\mathbf{p}}(\mathbf{x})\|_2\right)\right)^{n-1}(2\pi)^{-d/2}\exp\left(-\frac{1}{2}\|\mathbf{x}\|_2^2\right) \quad (4)$$

where $\|\text{proj}_{\mathbf{p}}(\mathbf{x})\|_2$ denotes the length of $\mathbf{x}$ projected onto $\mathbf{p}$, and $\Phi$ denotes the CDF of the standard normal distribution in 1D.

Proofs are in Appendix A.2. Theorem 2 states the probability density of the closest gallery sample $\mathbf{q}(\mathbf{p})$ in terms of the location of $\mathbf{p}$, and Corollary 2 interprets the density function by splitting it into a small variance and large variance regime. When $\|\mathbf{p}\|$ is small, the blue term in Eq.2 dominates and $\Pr[\mathbf{q}(\mathbf{p}) = \mathbf{x}]$ looks like a Dirac delta, see Fig. 6 Left. In other words, the nearest neighbor is likely to be in a small region; Eq. 3 states this formally. When $\|\mathbf{p}\|$ is large, the red term in Eq. 2 dominates and $\Pr[\mathbf{q}(\mathbf{p}) = \mathbf{x}]$ looks like a Gaussian with the same variance as the sample distribution in all directions except for $\mathbf{p}$, see Fig. 6 Middle Left. Clearly, this implies that a query that is far away from the gallery distribution is unlikely to belong to the same Voronoi cell as its nearest neighbor. We sketch a geometric argument for this implication using the boundary of Voronoi cells in Appendix A.4 but do not give a formal proof.

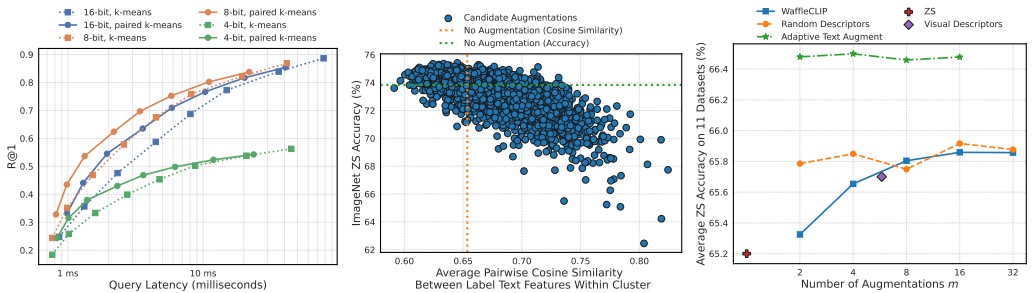

Figure 7: Left: Improvement of nearest neighbor recall with paired k-means at various latency settings and fine quantization levels. Middle: Correlation between variance of text features and ZS accuracy. Right: Intermediary ZS accuracy result with adaptive label augmentation.

---

**Algorithm 1** Paired k-means

1: **Input:** Image samples $\{\mathbf{x}_1, ..., \mathbf{x}_n\}$, text samples $\{\mathbf{p}_1, ..., \mathbf{p}_n\}$, number of clusters $k$.
2: Initialize cluster centroids $\{\mathbf{c}_1, ..., \mathbf{c}_k\}$.
3: Calculate the nearest image sample to each text sample, $\{\mathbf{q}(\mathbf{p}_1), ..., \mathbf{q}(\mathbf{p}_n)\}$. Note that there can be redundancies.
4: **for** a fixed number of iterations **do**
5:    **Assign** each image sample in $\{\mathbf{q}(\mathbf{p}_1), ..., \mathbf{q}(\mathbf{p}_n)\}$ to the nearest centroid.
6:    **Update** each centroid $\mathbf{c}$ in $\{\mathbf{c}_1, ..., \mathbf{c}_k\}$ to be the mean of text features paired with image features assigned to the cluster:

$$\mathbf{c} = \frac{1}{|\mathrm{Vor}(\mathbf{c})|} \sum_{\mathbf{p} \in \{\mathbf{p}_1, ..., \mathbf{p}_n\} | \mathbf{q}(\mathbf{p}) \in \mathrm{Vor}(\mathbf{c})} \mathbf{p}$$

7:    Normalize centroids.
8: **end for**
9: **Output:** cluster centroids $\{\mathbf{c}_1, ..., \mathbf{c}_k\}$.

---

**Algorithm 2** MUDG

**Input:** Source dataset $\mathcal{X}_s$, $\mathcal{A}_1 ... \mathcal{A}_m$, $n_{\mathrm{neighbors}}$, $k_1$, pretrained $f_{\mathrm{index}}$ and $f_{\mathrm{student}}$, $\{\mathbf{t}_1, ..., \mathbf{t}_c\}$.
**Step 1:** Let $\mathcal{Q} = \{f_{\mathrm{index, text}}(\mathcal{A}_j(\mathbf{t}_i)), \forall i = 1 : c, j = 1 : m\}$ denote the query set. For $q \in \mathcal{Q}$, retrieve $n_{\mathrm{neighbors}}$ closest samples in $\mathcal{X}_s$. Combine retrieved images from all queries; denote as $\mathcal{X}_1$.
**Step 2:** For $q \in \mathcal{Q}$, sort $\mathbf{x} \in \mathcal{X}_1$ by decreasing cosine similarity between $q$ and $f_{\mathrm{index, image}}(\mathbf{x})$. Denote rank of $\mathbf{x}$ relative to $q$ as $\mathrm{rank}(\mathbf{x}, q) \geq 1$. Assign each $\mathbf{x} \in \mathcal{X}_1$ the label corresponding to the closest ranked query, i.e. $\arg\min_q \mathrm{rank}(\mathbf{x}, q)$. Denote the labeled set as $\mathcal{X}_2$.
**Step 3:** Initialize an empty labeled dataset $\mathcal{X}_3$. For each label $y \in 1 : c$, find the subset of $\mathcal{X}_2$ with label $y$. Cluster into $k_1$ clusters, using k-means. Randomly select one sample from each cluster and append to $\mathcal{X}_3$. $\mathcal{X}_3$ contains $ck_1$ samples.
**Step 4:** Finetune $f_{\mathrm{student}}$ on $\mathcal{X}_3$ for $N$ iterations, using $\mathcal{L}_{\mathrm{train}}$ (Eq. 7).
**Output:** finetuned $f_{\mathrm{student}}$.

---

**Paired k-means** The fundamental issue causing the degradation in cross-modal recall is that the image centroids and queries are far away from each other in feature space. Consequently, the nearest centroid to a query does not provide much information about the location of the true nearest neighbor. To resolve this issue, we modify the k-means algorithm to update the centroids with the average of text features instead of image features. See Algorithm 1. This algorithm is an attempt at simultaneous minimization of the following two objectives heuristically:

$$\mathcal{L}_{\mathrm{kmeans}} = \frac{1}{n} \sum_{i=1}^{k} \sum_{\mathbf{x} \in \mathrm{Vor}(\mathbf{c}_i)} \|\mathbf{x} - \mathbf{c}_i\|_2^2, \quad \mathcal{L}_{\mathrm{cross\text{-}modal}} = \frac{1}{n} \sum_{\mathbf{p} \in \{\mathbf{p}_1, ..., \mathbf{p}_n\}} \mathbf{1}[\mathbf{c}(\mathbf{q}(\mathbf{p})) \neq \mathbf{c}(\mathbf{p})] \quad (5)$$

where $\{\mathbf{p}_1, ..., \mathbf{p}_n\} \in \mathcal{S}^d$ denotes a set of $n$ text queries, and $\mathbf{c}(\mathbf{p}) := \arg\min_{\mathbf{c}_i \sim \{\mathbf{c}_1, ..., \mathbf{c}_k\}} \|\mathbf{c}_i - \mathbf{p}\|_2$ denotes the closest centroid to a query $\mathbf{p}$. The first objective is the k-means objective. The second objective is the fraction of text queries $\mathbf{p}$ whose nearest centroid $\mathbf{c}(\mathbf{p})$ is different from the closest centroid to the nearest image sample $\mathbf{c}(\mathbf{q}(\mathbf{p}))$. The first objective enforces good clustering, while the second objective forces query features to be mapped to the same Voronoi cell as the nearest gallery feature. We show that both objectives converge empirically in Fig. 6 Right.

**Nearest neighbor search results** Figure 7 Left shows that an index trained with paired k-means outperforms the standard k-means index in R@1 for various values of $n_{\mathrm{probe}}$ and fine quantization levels. The cross-modal recall is directly related to the downstream target accuracy, since subsequent steps in our method rely on retrieving images that are relevant to the target task.

## 3.2 Diversified Retrieval with Adaptive Text Augmentation

There is very little semantic diversity among the nearest neighbors of any single query, which likely leads to severe overfitting during training, see Figures 16 and 17 in the Appendix. To ensure diversity of finetuning data, we propose to search the source dataset with augmented text queries in the format of "a photo of a ⟨label⟩, ⟨descriptor⟩." Previously, the authors of visual descriptors Menon and Vondrick (2022) proposed to use GPT to generate phrases that describe the label, e.g. "a photo of a chicken, which has two legs". Subsequently, waffleCLIP Roth et al. (2023) showed that the visual descriptors achieve similar zero-shot accuracies as random text augmentations on diverse datasets, see Figure 7 Right.

We consider two factors when choosing an appropriate augmentation function: (1) the augmented text does not change the label of the original text; and (2) the resulting distribution of augmented queries covers the entire concept of the class. The first requirement can be measured by the zero-shot accuracy of an ensemble of augmented texts. Let $\{\mathcal{A}_1, ..., \mathcal{A}_M\}$ denote a set of $M$ text augmentation functions. We aim to select a subset of size $m << M$ that does not change the meaning of the labels. We use the heuristic in Eq. 6 to choose the augmentation subset based on the target labels $\{\mathbf{t}_1, ..., \mathbf{t}_c\}$. First, we cluster the label text features into $k_2$ clusters using k-means. Denote the label clusters as $\{\mathcal{S}_{\mathbf{t},1}, ..., \mathcal{S}_{\mathbf{t},k_2}\}$ and the text encoder as $f_{\text{text}}$:

$$\underset{\mathcal{A} \sim \{\mathcal{A}_1,...,\mathcal{A}_M\}}{\arg\min} \sum_{i=1}^{k_2} \mathbf{1}\left[\sum_{\mathbf{t}_i,\mathbf{t}_j \sim \mathcal{S}_{\mathbf{t},i}} \langle f_{\text{text}}(\mathcal{A}(\mathbf{t}_i)), f_{\text{text}}(\mathcal{A}(\mathbf{t}_j)) \rangle > \sum_{\mathbf{t}_i,\mathbf{t}_j \sim \mathcal{S}_{\mathbf{t},i}} \langle f_{\text{text}}(\mathbf{t}_i), f_{\text{text}}(\mathbf{t}_j) \rangle \right] \quad (6)$$

Intuitively, an augmentation is desireable if it does not reduce the variance of the text features within any label cluster. Eq. 6 measures the variance of label features using their average pairwise cosine similarities, and counts the number of clusters where the augmentation $\mathcal{A}$ decreases this variance. For example, on ImageNet, the augmentation "a photo of a ⟨label⟩, which can be any size or shape" is a good augmentation because it does not reduce the distance between any two ImageNet labels, and the indicator function in Eq. 6 evaluates to 0 for all label clusters. On the other hand "a photo of a ⟨label⟩, which has sharp teeth" is a bad augmentation for ImageNet because it reduces the distance among text features corresponding to animal labels. This reduction degrades the model's ability to discriminate among the labels within the cluster, and the ZS accuracy decreases as a consequence, see Figure 7 Middle. Table 11 in the Appendix provides qualitative examples of augmentations with varying loss values.

We select the $m$ augmentations $\{\mathcal{A}_1, ..., \mathcal{A}_m\}$ with the lowest loss according to Eq. 6 and construct a dataset with $mc$ queries: $\{f_{\text{index,text}}(\mathcal{A}_j(\mathbf{t}_i)), \forall i = 1 : c, j = 1 : m\}$. $f_{\text{index,text}}$ denotes the text encoder used for indexing the source dataset $\mathcal{X}_s$. We retrieve the $n_{\text{neighbors}}$ nearest neighbors to each query in $\mathcal{X}_s$ and remove redundancies, resulting in a preliminary dataset size of at most $mcn_{\text{neighbors}}$. See step 1 of Algorithm 2.

## 3.3 Additional Tricks for Sample Selection and Finetuning

We label each retrieved image sample according to the text feature to which it is ranked the highest, see step 2 of Algorithm 2 and Appendix C.2 for a justification. We then select $k_1$ images for each label according to step 3 of Algorithm 2; the detailed procedure is presented in Appendix C.3. Finally, we finetune using the diversity preserving loss presented in Liao et al. (2023):

$$\mathcal{L}_{\text{train}} = \frac{1}{m} \sum_{\mathcal{A} \sim \{\mathcal{A}_1,...,\mathcal{A}_m\}} \text{CE}\left(\hat{y}_{\mathcal{A}}, (1 - \lambda)y + \lambda \hat{y}_{\mathcal{A},0}\right) \quad (7)$$

where CE denotes the cross entropy loss, $\hat{y}_{\mathcal{A}} \in \Delta_c$ denotes the soft prediction of the model with augmentation $\mathcal{A}$, $y$ denotes the one-hot encoded pseudo-label, and $\hat{y}_{\mathcal{A},0}$ denotes the soft prediction of the initial model with augmentation $\mathcal{A}$. $\lambda$ is a hyperparameter. $\mathcal{L}_{\text{train}}$ learns the pseudo-labels while simultaneously preserving the diversity present in the initial text encoder.

## 4 Experiments

We experiment with the ViT B/16 and ViT L/14 pretrained weights released by Radford et al. (2021) and available through the Python openclip package (Ilharco et al., 2021). The indexing model is ViT L/14; we modify FAISS (Douze et al., 2024) to build a search index for the source dataset, LAION-2B-en (Schuhmann et al., 2022). We experiment with two model sizes to show that we achieve large gains in target accuracies even when the indexing model and the student model are identical.

| | Setting | ImageNet | Caltech | Pets | Cars | Flowers | Food | Aircraft | SUN | DTD | EuroSAT | UCF | **Mean** |
|---|---|---|---|---|---|---|---|---|---|---|---|---|---|
| **Open-AI CLIP ViT-B/16** | | | | | | | | | | | | | |
| CLIP ZS (Radford et al., 2021) | ZS | 67.1 | 93.3 | 89.0 | 65.4 | 71.0 | 85.7 | 24.9 | 63.2 | 43.6 | 46.6 | 67.4 | 65.2 |
| waffleCLIP (Roth et al., 2023) | ZS | 68.2 | 93.5 | 88.1 | 65.5 | 72.1 | 85.9 | 25.6 | 66.2 | 44.3 | 47.3 | 68.1 | 65.9 |
| Random Desc. (Roth et al., 2023) | ZS | 68.1 | 94.3 | 87.7 | 65.7 | 71.7 | 85.7 | 25.2 | 66.2 | 44.7 | 47.7 | 67.3 | 65.8 |
| Ensemble (Radford et al., 2021) | ZS | 68.4 | 93.5 | 88.8 | 66.0 | 71.1 | 86.0 | 24.9 | 66.0 | 43.9 | 45.0 | 68.0 | 65.6 |
| Vis. Desc. (Menon and Vondrick, 2022) | ZS | 68.6 | 93.7 | 89.0 | 65.1 | 72.1 | 85.7 | 23.9 | 67.4 | 43.9 | 46.4 | 66.8 | 65.7 |
| CuPL Pratt et al. (2023) | ZS | 69.1 | - | 91.7 | 65.0 | 73.5 | 86.0 | 27.7 | 68.5 | 48.9 | - | 70.2 | - |
| SuS-X † Udandarao et al. (2023) | MUDG | 70.0 | 93.9 | 91.6 | 65.9 | 73.1 | 86.1 | 30.5 | 67.9 | **55.3** | 58.1 | 66.7 | 69.0 |
| Nearest neighbors | MUDG | 69.4 | 93.9 | **93.4** | 70.2 | 75.8 | 86.3 | 27.2 | 67.4 | 52.4 | 41.2 | 69.9 | 67.9 |
| Margin (Coleman et al., 2019) | MUDG | 64.5 | 94.7 | 92.8 | 61.6 | 75.5 | 86.3 | 33.2 | 61.6 | 51.0 | 59.6 | 71.9 | 68.4 |
| Least Confident (Coleman et al., 2019) | MUDG | 62.7 | 93.9 | 92.2 | 68.6 | 75.3 | 86.2 | **33.6** | 63.3 | 52.3 | 58.3 | 71.2 | 68.9 |
| Entropy (Coleman et al., 2019) | MUDG | 63.9 | 94.5 | 92.7 | 61.5 | 75.5 | 86.4 | 33.3 | 60.6 | 52.2 | 59.2 | 71.5 | 68.3 |
| Herding (Welling, 2009) | MUDG | 69.0 | **95.1** | 93.0 | **74.2** | 75.9 | 86.4 | 32.8 | 68.5 | 53.4 | 59.0 | **72.0** | 70.9 |
| K-Ctr Greedy (Sener and Savarese, 2017) | MUDG | 67.5 | 94.7 | 93.3 | 72.5 | 75.6 | 86.4 | 33.3 | 68.5 | 53.6 | 58.7 | 71.9 | 70.5 |
| **MUDG (ours)** | MUDG | **70.4** | 94.6 | 92.9 | 73.8 | **76.5** | **86.7** | 32.8 | **68.8** | 53.3 | **61.3** | 71.0 | **71.1** |
| Upper bound | | 73.9 | 95.8 | 95.1 | 89.9 | 95.9 | 87.5 | 59.2 | 77.3 | 73.2 | 89.0 | 86.4 | 83.9 |
| **Open-AI CLIP ViT-L/14** | | | | | | | | | | | | | |
| CLIP ZS (Radford et al., 2021) | ZS | 73.8 | 94.6 | 93.6 | 76.9 | 79.4 | 90.9 | 32.8 | 68.0 | 52.7 | 56.2 | 74.7 | 72.1 |
| waffleCLIP (Roth et al., 2023) | ZS | 75.0 | 96.1 | 93.5 | 77.1 | 78.8 | 90.9 | 33.6 | 69.3 | 54.3 | 57.7 | 75.3 | 72.9 |
| Random Desc. (Roth et al., 2023) | ZS | 75.1 | **96.9** | 93.4 | 76.7 | 78.5 | 90.7 | 33.6 | 70.1 | 54.5 | 59.3 | 75.5 | 73.1 |
| Ensemble (Radford et al., 2021) | ZS | 75.6 | 95.6 | 94.0 | 78.1 | 79.8 | 91.2 | 32.7 | 70.5 | 54.0 | 55.2 | 75.0 | 72.9 |
| Vis. Desc. (Menon and Vondrick, 2022) | ZS | 75.3 | 96.7 | 93.8 | 77.4 | 79.3 | 90.9 | 34.8 | 71.0 | 56.4 | 62.8 | 73.9 | 73.8 |
| CuPL Pratt et al. (2023) | ZS | 76.3 | - | 94.2 | 76.3 | 79.5 | 91.1 | **36.0** | 72.4 | **60.0** | - | 75.8 | - |
| Nearest neighbors | MUDG | 76.2 | 95.8 | **95.3** | 78.0 | **80.2** | 91.3 | 33.3 | 71.7 | 56.2 | 61.6 | 75.6 | 74.1 |
| **MUDG (ours)** | MUDG | **76.4** | 96.3 | 94.9 | **79.2** | 79.4 | 91.3 | 35.5 | **72.5** | 58.2 | 70.9 | 76.8 | **75.6** |

Table 2: Comparison of our MUDG baseline with ZS baselines and SuS-X on 11 diverse datasets. Average of three experiments. For MUDG rows, dataset construction and model training is separate for each dataset. "Nearest neighbors" refers to simple nearest neighbors retrieval. † indicates results reported by the authors; all other results are our reproductions. We finetune the ViT-B/16 model on 16-shot target training data as an upper bound.

| | | ImageNet | | | | | Office Home | | | | | DomainNet |
|---|---|---|---|---|---|---|---|---|---|---|---|---|
| | Setting | V2 | Sketch | A | R | **Mean** | A | C | P | R | **Mean** | **Mean** |
| **Open-AI CLIP ViT-B/16** | | | | | | | | | | | | |
| CLIP ZS Radford et al. (2021) | ZS | 60.9 | 46.6 | 47.2 | 74.1 | 57.2 | 82.6 | 67.2 | 88.8 | 89.6 | 82.1 | 57.6 |
| waffleCLIP Roth et al. (2023) | ZS | 61.8 | 48.5 | 50.0 | 76.3 | 59.2 | 83.1 | 68.2 | 89.7 | 90.4 | 82.9 | 59.7 |
| Random Desc. Roth et al. (2023) | ZS | 61.7 | 48.8 | 49.9 | 76.6 | 59.2 | 83.0 | 69.1 | 89.5 | 90.2 | 83.0 | 59.6 |
| Ensemble Radford et al. (2021) | ZS | 61.9 | 48.5 | 49.2 | 77.9 | 59.4 | 84.3 | 67.7 | 89.3 | 90.2 | 82.9 | 60.2 |
| Vis. Desc. Menon and Vondrick (2022) | ZS | 61.8 | 48.1 | 48.6 | 75.2 | 58.4 | - | - | - | - | - | - |
| PromptStyler † Cho et al. (2023) | SFDG | - | - | - | - | - | 83.8 | 68.2 | 91.6 | 90.7 | 83.6 | 59.4 |
| **MUDG (ours)** | MUDG | **63.6** | **50.4** | **51.5** | **80.1** | **61.4** | **85.9** | **73.3** | **92.0** | **91.4** | **85.7** | **61.2** |
| **Open-AI CLIP ViT-L/14** | | | | | | | | | | | | |
| CLIP ZS Radford et al. (2021) | ZS | 68.0 | 57.9 | 68.3 | 85.5 | 69.9 | 87.1 | 74.8 | 93.1 | 93.4 | 87.1 | 63.9 |
| waffleCLIP (Roth et al., 2023) | ZS | 68.8 | 58.7 | 70.1 | 87.1 | 71.2 | 87.7 | 78.2 | 93.8 | 94.4 | 88.5 | 65.4 |
| Random Desc. Roth et al. (2023) | ZS | 69.2 | 59.1 | 70.5 | 87.1 | 71.5 | 88.2 | 78.4 | 94.4 | 94.0 | 88.7 | 65.7 |
| Ensemble Radford et al. (2021) | ZS | 69.9 | 59.7 | 70.2 | 87.8 | 71.9 | 88.5 | 76.9 | 93.8 | 94.5 | 88.4 | 66.1 |
| Vis. Desc. Menon and Vondrick (2022) | ZS | 69.4 | 58.8 | 69.6 | 86.4 | 71.1 | - | - | - | - | - | - |
| PromptStyler † Cho et al. (2023) | SFDG | - | - | - | - | - | 89.1 | 77.6 | 94.8 | **94.8** | 89.1 | 65.5 |
| **MUDG (ours)** | MUDG | **70.1** | **60.9** | **72.1** | **89.0** | **73.0** | **90.2** | **81.5** | **95.1** | 94.6 | **90.3** | **67.0** |

Table 3: Comparison of our MUDG baseline with ZS baselines and PromptStyler on DG benchmarks. Average of three trials. Dataset construction and model training is performed once and evaluated on all domains for Office Home, Terra Incognita, and DomainNet; but we perform the steps separately for each ImageNet domain, due to differences in label spaces. PromptStyler (Cho et al., 2023) † results are those reported by the authors; all other results are our reproductions.

**Datasets** We experiment with a diverse set of target classification tasks. ImageNet-1K (Russakovsky et al., 2015), Caltech-101 (Li et al., 2022a), Oxford-Pets (Parkhi et al., 2012), Stanford-Cars (Krause et al., 2013), Flowers-102 (Nilsback and Zisserman, 2008), Food-101 (Bossard et al., 2014), FGVC-Aircraft (Maji et al., 2013), SUN-397 (Xiao et al., 2010), Describable-Textures (DTD) (Cimpoi et al., 2013), EuroSAT (Helber et al., 2019), UCF-101 (an action recognition dataset) (Soomro et al., 2012) in Table 2 and ImageNet-V2 (Recht et al., 2019), ImageNet-Sketch (Wang et al., 2019), ImageNet-A (natural adversarial examples) (Hendrycks et al., 2021b), and ImageNet-R (Hendrycks et al., 2021a) in Table 3 are commonly used by zero-shot papers, while Office Home (Venkateswara et al., 2017), Terra Incognita (Beery et al., 2018), DomainNet (Peng et al., 2019), VLCS (Torralba and Efros,

Figure 8: Summary of ablation experiments. See Appendix Tables 4, 5 and 6 for detailed tables.

2011), and PACS (Li et al., 2017a) are common DG and DA datasets. TerraInc, VLCS and PACS results are in Appendix B.

**Baselines** We compare to ZS (Roth et al., 2023; Radford et al., 2021; Menon and Vondrick, 2022; Pratt et al., 2023), SFDG (Cho et al., 2023), and MUDG (Udandarao et al., 2023) baselines. We also include a strong random descriptor baseline which ensembles randomly selected visual descriptors (Roth et al., 2023). To the best of our knowledge, PromptStyler (Cho et al., 2023) is the only current SFDG baseline and SuS-X (Udandarao et al., 2023) is the most suitable MUDG baseline. In addition, we compare against representative coreset selection baselines (Coleman et al., 2019; Welling, 2009; Sener and Savarese, 2017), using the implementations by Guo et al. (2022). We do not compare against supervised DG baselines, such as ERM and MIRO (Cha et al., 2022), since those methods require labeled data for the target task. **Ablations** We provide ablation studies justifying our paired k-means indexing, adaptive label augmentation, diversity preserving loss, and sample selection schemes in Tables 4, 5 and 6 in the Appendix and summarized in Fig. 8. **Hyperparameters** are listed in Tables 9 and 10 of the Appendix. An ablation study on $m$, $n_{neighbors}$, $k_1$ and $n_{probe}$ is included in Figure 15 of the Appendix. **Qualitative results** We provide qualitative examples of successful and unsuccessful retrievals in Figures 9 and 10.

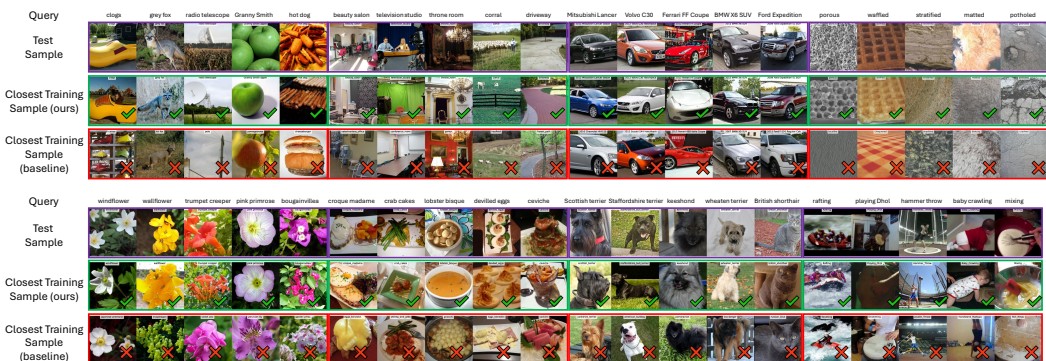

Figure 9: Qualitative comparison of our retrieval results against the baseline method (Udandarao et al., 2023). Our method retrieves images which are more aligned with the target concepts. Dataset names in order: ImageNet, SUN, Cars, DTD, Flowers, Food, Pets, UCF.

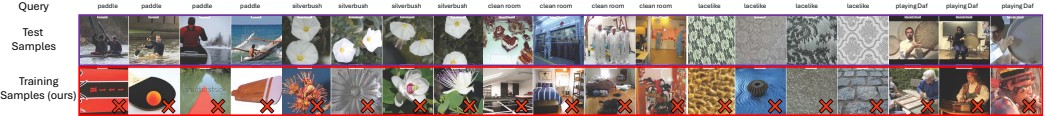

Figure 10: Example retrieval failure modes. Target concepts of "clean room" and "paddle" are different from the CLIP alignment; the CLIP embedding space is not aligned for the action "playing Daf", the texture "lacelike" and the flower "silverbush".

## 5 CONCLUSION

This work tackled the multimodal unsupervised domain generalization problem, which finetunes a model for a target task using images retrieved from a non-task-specific, unlabeled source dataset. We broke the MUDG problem down into three smaller sub-problems and proposed novel solutions for each sub-problem. First, we introduced a paired k-means clustering approach to build an index with superior cross-modal recall. Second, we designed an unsupervised heuristic to select good label augmentations for diversified retrieval. Finally, we trained the student CLIP model on the retrieved data with a diversity preserving loss to yield promising accuracy improvements across 20 diverse benchmarks.

ACKNOWLEDGEMENTS

DISTRIBUTION STATEMENT A. Approved for public release. Distribution is unlimited.

This material is based upon work supported by the Under Secretary of Defense for Research and Engineering under Air Force Contract No. FA8702-15-D-0001. Any opinions, findings, conclusions or recommendations expressed in this material are those of the author(s) and do not necessarily reflect the views of the Under Secretary of Defense for Research and Engineering.

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

LIMITATIONS

Even though we do not explicitly assume any relationships between the source and target data, *our work may not be applicable to problems where the target visual concepts are either not present in the source dataset or completely misaligned with corresponding language concepts*, see Figure 10. Possible examples include synthetic aperture radar images, images of tissue samples, or medical scans. Additionally, our method may not improve results on datasets where the zero-shot accuracy is already saturated. For example, the VLCS dataset contains 5 classes: bird, car, chair, dog, and person. Our method does not achieve any meaningful improvement over the ZS baseline on these simple classification tasks.

## A  PROOFS

### A.1  PROOF OF THEOREM 1

*Step 1.* For ease of notation, use $\mathbf{c}$ to denote closest centroid to $\mathbf{p}$, and use $\mathbf{q}$ to denote the closest point to $\mathbf{p}$:

$$\mathbf{q} := \underset{\mathbf{x}_i \sim \{\mathbf{x}_1, \dots, \mathbf{x}_n\}}{\arg\min} \|\mathbf{x}_i - \mathbf{p}\|_2 \tag{8}$$

First, we need to solve for the CDF of the probability distribution over the cosine similarity between $\mathbf{p}$ and $\mathbf{q}$.

$$\begin{aligned} \Pr[\langle \mathbf{p}, \mathbf{q} \rangle \geq s] &= 1 - \Pr[\langle \mathbf{p}, \mathbf{q} \rangle < s] \\ &= 1 - \Pi_{\mathbf{x}_i}^n \Pr[\langle \mathbf{p}, \mathbf{x}_i \rangle < s] \\ &= 1 - \Pi_{\mathbf{x}_i}^n (1 - \Pr[\langle \mathbf{p}, \mathbf{x}_i \rangle \geq s]) \end{aligned} \tag{9}$$

For ease of analysis, we assumed that $\text{Vor}(\mathbf{c})$ is a strict subset of the hemisphere centered at $\mathbf{c}$, so we only need to consider $s < 0$. This corresponds to $\theta < \pi/2$, where $\theta$ denotes the angle between $\mathbf{p}$ and $\mathbf{q}$. Since the $\mathbf{x}_i$s are independently uniformly distributed over $\mathcal{S}^d$, $\Pr[\langle \mathbf{p}, \mathbf{x}_i \rangle \geq s]$ in Eq. 9 corresponds to the ratio of the surface area of a spherical cap with angle $\theta = \cos^{-1}(s)$ to the entire surface area of the sphere. This ratio is given in Li (2010) Li (2010):

$$\Pr[\langle \mathbf{p}, \mathbf{x}_i \rangle \geq \cos \theta] = \frac{1}{2} I_{\sin^2 \theta} \left( \frac{d-1}{2}, \frac{1}{2} \right) := \rho(\cos(\theta)) \tag{10}$$

where $I \in [0, 1)$ is the regularized incomplete beta function. We will use $\rho \in [0, 0.5)$ to denote the surface area of a spherical cap as a function of the cosine similarity as a fraction of the surface area of $\mathcal{S}^d$.

Given $\mathbf{c}$, Pick $s'$ to be the closest point on the boundary to the Voronoi cell to $\mathbf{c}$, i.e. cosine of half the angle to the closest centroid:

$$s' := \cos \left( \frac{1}{2} \cos^{-1} \max_{\mathbf{c}_i \in \{\mathbf{c}_1, \dots, \mathbf{c}_k\} \setminus \mathbf{c}} \langle \mathbf{c}, \mathbf{c}_i \rangle \right) \tag{11}$$

Note that $s'$ is chosen such that the the spherical cap with $\theta = \cos^{-1}(s')$ is the largest possible spherical cap centered at $\mathbf{c}$ that is still fully contained within $\text{Vor}(\mathbf{c})$.

The probability in Eq. 1 can then be decomposed as:

$$\begin{aligned} \Pr[\mathbf{q} \in \text{Vor}(\mathbf{c})] = \underbrace{\Pr[\langle \mathbf{p}, \mathbf{q} \rangle \geq s'] \Pr[\mathbf{q} \in \text{Vor}(\mathbf{c}) \mid \langle \mathbf{p}, \mathbf{q} \rangle \geq s']}_{g_{\mathbf{c}}(\mathbf{p})} \\ + \underbrace{\Pr[\langle \mathbf{p}, \mathbf{q} \rangle < s']}_{\epsilon} \Pr[\mathbf{q} \in \text{Vor}(\mathbf{c}) \mid \langle \mathbf{p}, \mathbf{q} \rangle < s'] \end{aligned} \tag{12}$$

In the above equation, we hope that $n$ is large enough and $k$ is small enough such that the second term is small, and the theorem is only meaningful in this regime. Intuitively, a large $n$ leads to a exponentially diminishing probability that $\mathbf{q}$ is far away from $\mathbf{p}$, see Eq. 9; and a relatively small $k$ ensures that $\Pr[\langle \mathbf{p}, \mathbf{q} \rangle \geq s']$ is large. Let's denote $\epsilon := \Pr[\langle \mathbf{p}, \mathbf{q} \rangle < s']$, such that the second term in Eq. 12 can be bounded by 0 and $\epsilon$. This simplifies the analysis, since we now only need to

worry about what happens inside the spherical cap with angle $\cos^{-1}(s')$ around $\mathbf{p}$. By construction, $\Pr[\mathbf{q} \in \mathrm{Vor}(\mathbf{c}) \mid \langle \mathbf{c}, \mathbf{q} \rangle \geq s'] = 1$, so $g_{\mathbf{c}}(\mathbf{c}) = \Pr[\langle \mathbf{p}, \mathbf{q} \rangle \geq s'] = \rho(s')$. This is property 1 of Theorem 1.

*Step 2.* The proof of property 3 of Theorem 1 (monotonicity of $g_{\mathbf{c}}$) can be proven from the convexity of Voronoi cells. $g_{\mathbf{c}}$ from Eq. 12 can be written as an integral over a spherical cap of probability density multiplied by an indicator function of whether that part of the spherical cap is still within the Voronoi cell. We can establish monotonicity of each indicator function by simply noticing that a ray originating from a point strictly within a convex set can only cross the boundary of that convex set once.

Let $\mathcal{C}^{\theta}(\mathbf{p})$ denote the spherical cap in $\mathcal{S}^d$ centered around $\mathbf{p}$ with $\theta = \cos^{-1}(s')$. Then,

$$g_{\mathbf{c}}(\mathbf{p}) = (1 - \epsilon) \int_{\mathbf{v} \in \mathcal{C}^{\theta}(\mathbf{p})} \Pr[\mathbf{v} = \mathbf{q} \mid \mathbf{q} \in \mathcal{C}^{\theta}(\mathbf{p})] \mathbf{1}[\mathbf{v} \in \mathrm{Vor}(\mathbf{c})] d\mathbf{v} \tag{13}$$

where $\Pr[\mathbf{v} = \mathbf{q}]$ denotes the probability density function that $\mathbf{v}$ is the closest sample to $\mathbf{p}$ (out of the $n$ samples). $\mathbf{1}$ is the indicator function. When $\mathbf{p} = \mathbf{c}$, all the indicator functions in Eq. 13 are equal to 1. All indicator functions are non-increasing in all directions from within the Voronoi cell in the following sense:

$$\mathbf{1}\left[\mathrm{proj}_{\mathcal{S}^d}(a\mathbf{u} + \mathbf{v}) \in \mathrm{Vor}(\mathbf{c})\right] \leq \mathbf{1}\left[\mathrm{proj}_{\mathcal{S}^d}(b\mathbf{u} + \mathbf{v}) \in \mathrm{Vor}(\mathbf{c})\right], \forall a > b > 0, \mathbf{u} \in \mathbb{R}^d, \mathbf{v} \in \mathrm{Vor}(\mathbf{c}) \tag{14}$$

Eq. 14 follows from the convexity of $\mathrm{Vor}(\mathbf{c})$, and property 3 of Theorem 1 follows from the combination of Eq. 13 and 14.

*Step 3.* Finally, we conclude by showing property 2 of Theorem 1. This property states that $\Pr[\mathbf{q} \in \mathrm{Vor}(\mathbf{c}(\mathbf{b}))] \leq 0.5$ for all points $\mathbf{b}$ on the boundary of $\mathrm{Vor}(\mathbf{c})$. This is easy to see. We assumed that $\mathrm{Vor}(\mathbf{c})$ is a strict subset of a hemisphere. For any point $\mathbf{b}$, it is obviously possible to construct a hemisphere $\mathcal{C}^{\pi/2}$ such that all of $\mathrm{Vor}(\mathbf{c})$ is contained within $\mathcal{C}^{\pi/2}$ and $\mathbf{b}$ is on the boundary of $\mathcal{C}^{\pi/2}$. Clearly, the function $\Pr[\mathbf{q} \in \mathcal{C}^{\pi/2}]$ is symmetric around the boundary of the hemisphere $\mathcal{C}^{\pi/2}$, so $\Pr[\mathbf{q} \in \mathrm{Vor}(\mathbf{c}(\mathbf{b}))] \leq \Pr[\mathbf{q} \in \mathcal{C}^{\pi/2}] = 0.5$, since $\mathrm{Vor}(\mathbf{c}(\mathbf{b})) \subseteq \mathcal{C}^{\pi/2}$. $\square$

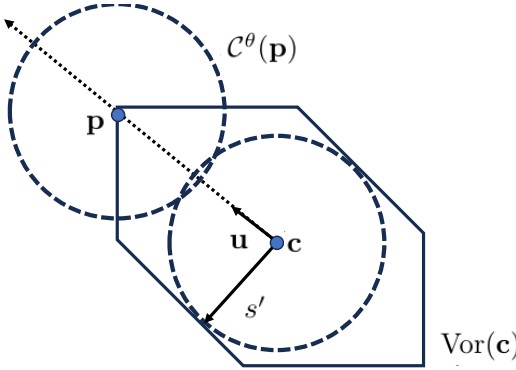

Figure 11: Diagram for proof of Theorem 1.

## A.2 PROOF OF THEOREM 2

Firstly, we want to derive the probability that the closest point to $\mathbf{p}$ lies in $\mathcal{B}_r(\mathbf{p})$:

$$\begin{aligned} \Pr[\mathbf{q}(\mathbf{p}) \in \mathcal{B}_r(\mathbf{p})] &= 1 - \Pr[\mathbf{q}(\mathbf{p}) \notin \mathcal{B}_r(\mathbf{p})] \\ &= 1 - \Pr[\mathbf{x}_i \notin \mathcal{B}_r(\mathbf{p})]^n \\ &= 1 - (1 - \Pr[\mathbf{x}_i \in \mathcal{B}_r(\mathbf{p})])^n \end{aligned} \tag{15}$$

Equation 15 can be considered a CDF with respect to the radius. Let's derive the PDF w.r.t. the radius by differentiating:

$$\begin{aligned}
\Pr[\mathbf{q}(\mathbf{p}) \in \mathcal{S}_r(\mathbf{p})] &= \frac{d}{dr} \Pr[\mathbf{q}(\mathbf{p}) \in \mathcal{B}_r(\mathbf{p})] \\
&= \frac{d}{dr} \left[ 1 - (1 - \Pr[\mathbf{x}_i \in \mathcal{B}_r(\mathbf{p})])^n \right] \\
&= -\frac{d}{dr} (1 - \Pr[\mathbf{x}_i \in \mathcal{B}_r(\mathbf{p})])^n \\
&= n \left(1 - \Pr[\mathbf{x}_i \in \mathcal{B}_r(\mathbf{p})]\right)^{n-1} \frac{d}{dr} \Pr[\mathbf{x}_i \in \mathcal{B}_r(\mathbf{p})]
\end{aligned} \tag{16}$$

Let $K = (2\pi)^{-d/2}$. Let $\Pr[\mathbf{x}_i \in \mathcal{B}_r(\mathbf{p})]$ denote the probability that a single point drawn from $\mathcal{N}(\mathbf{0}, \mathbf{I}_d)$ resides in the $\mathbb{R}^d$ ball of radius $r$ centered at $\mathbf{p}$. $\mathcal{S}_r(\mathbf{p})$ denotes the $\mathbb{R}^d$ sphere of radius $r$ around $\mathbf{p}$ (the boundary of $\mathcal{B}_r(\mathbf{p})$).

$$\begin{aligned}
\Pr[\mathbf{x}_i \in \mathcal{B}_r(\mathbf{p})] &= \int_{\mathbf{v} \in \mathcal{B}_r(\mathbf{p})} K \exp\left(-\frac{1}{2}\|\mathbf{v}\|_2^2\right) d\mathbf{v} \\
&= \int_{r'=0}^{r'=r} \int_{\mathbf{v} \in \mathcal{S}_r(\mathbf{p})} K \exp\left(-\frac{1}{2}\|\mathbf{v}\|_2^2\right) d\mathbf{v}\, dr' \\
\frac{d}{dr} \Pr[\mathbf{x}_i \in \mathcal{B}_r(\mathbf{p})] &= \int_{\mathbf{v} \in \mathcal{S}_r(\mathbf{p})} K \exp\left(-\frac{1}{2}\|\mathbf{v}\|_2^2\right) d\mathbf{v}
\end{aligned} \tag{17}$$

Substitute Eq. 17 into 16 to get the probability density that the closest sample to $\mathbf{p}$ lies in $\mathcal{S}_r(\mathbf{p})$. Note that the fact that $\mathbf{q}(\mathbf{p})$ is the closest point to $\mathbf{p}$ does not change the marginal distribution w.r.t. $r$, so $\Pr[\mathbf{x}_i = \mathbf{x} \mid \mathbf{x}_i \in \mathcal{S}_r(\mathbf{p})] = \Pr[\mathbf{q}(\mathbf{p}) = \mathbf{x} \mid \mathbf{q}(\mathbf{p}) \in \mathcal{S}_r(\mathbf{p})]$.

$$\Pr[\mathbf{q}(\mathbf{p}) = \mathbf{x} \mid \mathbf{q}(\mathbf{p}) \in \mathcal{S}_r(\mathbf{p})] = \frac{K \exp\left(-\frac{1}{2}\|\mathbf{x}\|_2^2\right)}{\int_{\mathbf{v} \in \mathcal{S}_r(\mathbf{p})} K \exp\left(-\frac{1}{2}\|\mathbf{v}\|_2^2\right) d\mathbf{v}}, \ \forall \mathbf{x} \in \mathcal{S}_r(\mathbf{p}) \tag{18}$$

When we substitute, the integrals cancel out:

$$\begin{aligned}
\Pr[\mathbf{q}(\mathbf{p}) = \mathbf{x}] &= \Pr[\mathbf{q}(\mathbf{p}) = \mathbf{x} \mid \mathbf{q}(\mathbf{p}) \in \mathcal{S}_r(\mathbf{p})] \Pr[\mathbf{q}(\mathbf{p}) \in \mathcal{S}_r(\mathbf{p})] \\
&= n \left(1 - \Pr[\mathbf{x}_i \in \mathcal{B}_r(\mathbf{p})]\right)^{n-1} (2\pi)^{-d/2} \exp\left(-\frac{1}{2}\|\mathbf{x}\|_2^2\right), \ r := \|\mathbf{x} - \mathbf{p}\|_2, \ \forall \mathbf{x} \in \mathbb{R}^d
\end{aligned} \tag{19}$$

$\square$

### A.3 PROOF OF COROLLARY 2

*Part 1.* This part is straightforward.

$$\begin{aligned}
\Pr[\|\mathbf{q}(\mathbf{p}) - \mathbf{p}\| > r] &= (1 - \Pr[\mathbf{x}_i \in \mathcal{B}_r(\mathbf{p})])^n \\
&< \left(1 - \left(V_r K \exp\left(-\frac{1}{2}(\|\mathbf{p}\| + r)^2\right)\right)\right)^n
\end{aligned} \tag{20}$$

where $V_r = \frac{\pi^{d/2}}{\Gamma(d/2+1)}$ is the volume of a ball of radius $r$ in $\mathbb{R}^d$. $K = (2\pi)^{-d/2}$. The upper bound in Eq. 20 comes from lower bounding the probability density of $\mathbf{x}_i \sim \mathcal{N}(\mathbf{0}, \mathbf{I}_d)$ within $\mathcal{B}_r(\mathbf{p})$ with the smallest value.

*Part 2.* WLOG assume $\mathbf{p}$ lies along the first coordinate axis; let the scalar value of this coordinate be $p := \|\mathbf{p}\|$ for simplicity. Let's introduce a constant $k \in (0, 1)$. Consider the probability that the first coordinate of the $n$ samples is greater than $p^k$. This is the probability that the max of $n$ independent samples $\{x_1, ..., x_n\} \sim \mathcal{N}(0, 1)$ is bigger than $p^k$. This is a standard result using a Chernoff-derived tail bound and a union bound (Vershynin, 2018):

$$\Pr\left[\max(x_1, ..., x_n) \leq \sqrt{2 \ln \frac{n}{\delta}}\right] \geq 1 - \delta, \ \delta \in (0, 1) \tag{21}$$

Using our value of $p^k$ for the bound, we get:

$$\Pr\left[\max(x_1, ..., x_n) \leq p^k\right] \geq 1 - ne^{-p^{2k}/2} \tag{22}$$

Equation 22 implies that $\Pr[\mathbf{q}(\mathbf{p})[0] > p^k] \to 0$ as $p \to \infty$, for $k \in (0, 1)$, where $\mathbf{q}(\mathbf{p})[0]$ denotes the first coordinate of $\mathbf{q}$. Using another union bound, we can easily show that as $p \to \infty$, the probability that $\mathbf{q}(\mathbf{p})$ resides within a hypercube with side lengths $2p^k$ goes to 1, exponentially. Concretely, let $\|\cdot\|_\infty$ denote the infinity norm, then:

$$\Pr\left[\max(\|\mathbf{x}_1\|_\infty, ..., \|\mathbf{x}_n\|_\infty) \leq p^k\right] \geq 1 - nde^{-p^{2k}/2} \tag{23}$$

Now, we only need to consider the probability mass within this hypercube. We consider the approximation of the ball $\mathcal{B}_{\|\mathbf{x}-\mathbf{p}\|}(\mathbf{p})$ with the half-space $\mathcal{H}_{\mathbf{x}[0]} := \{\mathbf{x}' \in \mathbb{R}^d \mid \mathbf{x}'[0] \geq \mathbf{x}[0]\}$. Considering only points that lie within the hypercube with side lengths $2p^k$, the difference in between the union and intersection of $\mathcal{B}_{\|\mathbf{x}-\mathbf{p}\|}(\mathbf{p})$ and $\mathcal{H}_{\mathbf{x}[0]}$ goes to zero. This is because the discrepancy between the two sets is a spherical cap with max height $h = \sqrt{(p - p^k)^2 + (d-1)p^{2k}} - (p - p^k)$. This occurs at the corner of the hypercube. As $p \to \infty$, $h \to 0$, for $k < \frac{1}{2}$. This limit can be easily seen by writing $h$ as a fraction:

$$h = \frac{a^2 - b^2}{a + b} = \frac{(d-1)p^{2k}}{\Theta(p + \sqrt{d}p^k)} \ , \ a := \sqrt{(p - p^k)^2 + (d-1)p^{2k}} \ , b := p - p^k$$

Therefore, $\Pr[\mathbf{x}_i \in \mathcal{B}_{\|\mathbf{x}-\mathbf{p}\|}(\mathbf{p})] \to \Pr[\mathbf{x}_i[0] \geq \mathbf{x}[0]]$. The later probability is the tail of a 1D Gaussian, $1 - \Phi(\mathbf{x}[0])$. Substituting this into Eq. 2 of Theorem 2, we recover the limiting distribution in Eq. 4 of the corollary. $\square$

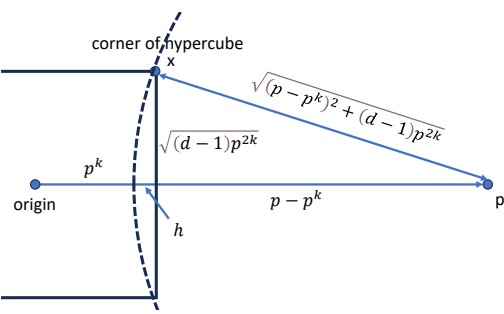

Figure 12: Diagram for proof of Corollary 2.

### A.4 ALTERNATIVE GEOMETRIC INTUITION TO THEOREM 2

As an alternative intuition to Theorem 2, consider the set difference between the Voronoi cell defined by centroids and the union of all Voronoi cells defined by individual gallery samples within the cluster. Queries that fall into this difference region do not retrieve the correct nearest neighbor. This difference region grows as a query moves farther away from the gallery distribution. We illustrate this intuition in Figure 13. For this figure, we generate 10,000 2D Gaussian samples and cluster them into 20 clusters. The Voronoi cells of the 20 clusters is plotted in solid black lines. The Voronoi cells formed by the 10,000 samples are also plotted and color-coded by cluster. When a query belongs a Voronoi cell that is different from the Voronoi cell of the closest sample, nearest neighbor retrieval fails. Clearly, the approximation of the union of Voronoi cells of samples by the Voronoi regions of the centroids becomes worse with increasing distance from the origin. This results in lower retrieval accuracy.

## B EXPERIMENTAL DETAILS AND ADDITIONAL RESULTS

**Domain abbreviations**  Office Home domains: A - art; C - clipart; P - product; R - real. Terra Incognita domain names are anonymous location identifiers for camera traps. DomainNet domains: C - clipart; I - infograph; P - painting; Q - quickdraw; R - real; S - sketch. PACS domains: A - art; C - cartoon; P - photo; S - sketch. VLCS domain names are dataset names.

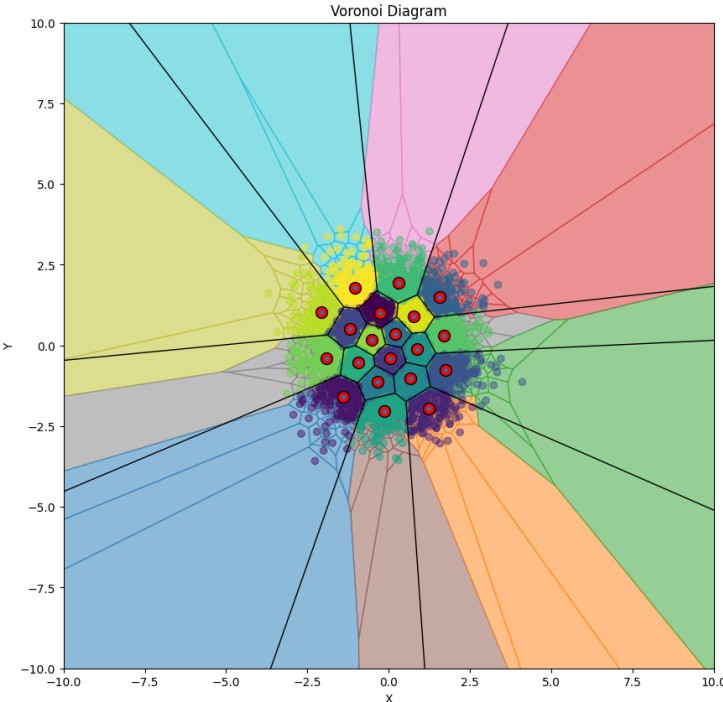

Figure 13: Alternative intuition to Theorem 2 in Section A.4. We generate 10,000 2D Gaussian samples and assign them to 20 clusters. The Voronoi cells of the 20 clusters is plotted in solid black lines. The Voronoi cells formed by the 10,000 samples are also plotted and color-coded by cluster. When a query belongs a Voronoi cell that is different from the Voronoi cell of the closest sample, retrieval fails. Clearly, the approximation of the union of Voronoi cells of samples by the Voronoi regions of the centroids becomes worse with increasing distance from the origin.

| Paired k-means | Adaptive augmentation | Diversity preserving loss | 11 datasets | | | | | | | | | | | | ImageNet | | | | |
|---|---|---|---|---|---|---|---|---|---|---|---|---|---|---|---|---|---|---|---|
| | | | ImageNet | Caltech | Pets | Cars | Flowers | Food | Aircraft | SUN | DTD | EuroSAT | UCF | Mean | V2 | Sketch | A | R | Mean |
| | | | | | | | | | Open-AI CLIP ViT-B/16 | | | | | | | | | | |
| ✓ | | ✓ | 70.3 | **94.6** | **93.5** | 72.7 | 75.9 | 86.6 | **33.9** | 69.1 | **54.7** | 59.1 | 71.5 | **71.1** | 63.5 | 50.0 | **51.9** | 79.0 | 61.1 |
| | ✓ | | 70.0 | **94.6** | **93.5** | 74.6 | 75.7 | 86.5 | 33.3 | 69.5 | 53.7 | 57.0 | 69.2 | 70.7 | 63.0 | 49.9 | 49.1 | 79.1 | 60.2 |
| | ✓ | ✓ | **70.4** | **94.6** | 93.4 | 73.4 | 75.3 | 86.4 | 32.9 | **69.6** | 54.0 | 58.9 | 69.0 | 70.7 | 63.5 | 50.2 | 51.3 | 79.4 | 61.1 |
| ✓ | ✓ | | 70.2 | 94.3 | 93.2 | **75.2** | 76.4 | 86.5 | 33.5 | 68.5 | 53.1 | 59.5 | **71.9** | **71.1** | 62.9 | 50.1 | 49.8 | 79.7 | 60.6 |
| ✓ | ✓ | ✓ | **70.4** | **94.6** | 92.9 | 73.8 | **76.5** | **86.7** | 32.8 | 68.8 | 53.3 | **61.3** | 71.0 | **71.1** | **63.6** | **50.4** | 51.5 | **80.1** | **61.4** |

Table 4: Ablations experiments part 1.

**Hyperparameters**    The finetuning parameters are displayed in Table 9. The training set construction parameters $n_{\text{neighbors}}$, $m$, and $k_1$ are dataset-specific and listed in Table 10. Moreover, the number of training iterations $N$ and query/prompt template also varies with the dataset, as listed in Table 10.

**Ablation study**    An ablation study on the training set construction parameters $n_{\text{neighbors}}$, $m$, $k_1$ and $n_{\text{probe}}$ are included in Figure 15. We perform these experiments for ImageNet, DomainNet, and Office Home. When varying the values of $n_{\text{neighbors}}$ and $m$, we scale the value of $k_1$ by the same amount. Note that changing the values of these hyperparameters changes the size of the training

| Paired k-means | Adaptive augmentation | Diversity preserving loss | DomainNet | | | | | | | OfficeHome | | | | |
|---|---|---|---|---|---|---|---|---|---|---|---|---|---|---|
| | | | C | I | P | Q | R | S | **Mean** | A | C | P | R | **Mean** |
| | | | | | | Open-AI CLIP ViT-B/16 | | | | | | | | |
| ✓ | | ✓ | 74.8 | **54.3** | 69.1 | 15.5 | 85.5 | 66.3 | 60.9 | 85.5 | 73.1 | **92.0** | **91.4** | 85.5 |
| | ✓ | | 74.8 | 52.6 | 68.9 | 15.7 | 85.2 | 66.1 | 60.5 | 85.2 | 70.7 | 91.0 | 90.5 | 84.3 |
| | ✓ | ✓ | 74.9 | 53.7 | 69.5 | 16.1 | 85.4 | 66.3 | 61.0 | 84.8 | 70.4 | 90.9 | 90.9 | 84.3 |
| ✓ | ✓ | | **75.3** | 52.6 | 69.6 | 16.3 | 85.4 | 66.5 | 60.9 | 85.8 | 72.7 | 91.9 | 90.9 | 85.3 |
| ✓ | ✓ | ✓ | **75.3** | 53.8 | **69.8** | **16.4** | **85.6** | **66.6** | **61.2** | **85.9** | **73.3** | **92.0** | **91.4** | **85.7** |

Table 5: Ablation experiments part 2.

| | ImageNet | Caltech | Pets | Cars | Flowers | Food | Aircraft | SUN | DTD | EuroSAT | UCF | **Mean** |
|---|---|---|---|---|---|---|---|---|---|---|---|---|
| | | | | | Open-AI CLIP ViT-B/16 | | | | | | | |
| Nearest neighbors | 69.4 | 93.9 | 93.4 | 70.2 | 75.8 | 86.3 | 27.2 | 67.4 | 52.4 | 41.2 | 69.9 | 67.9 |
| Soft pseudo labels | 69.9 | **94.8** | 93.1 | 70.7 | 74.7 | **86.7** | 31.9 | 67.6 | 52.3 | 51.5 | 70.8 | 69.5 |
| Contrastive loss (Khosla et al., 2020) | 68.4 | 93.7 | 93.4 | **72.7** | **77.0** | 86.3 | 33.7 | 68.6 | 54.3 | **59.8** | **71.6** | 70.9 |
| Our training loss | **70.3** | 94.6 | **93.5** | 72.7 | 75.9 | 86.6 | **33.9** | **69.1** | **54.7** | 59.1 | 71.5 | **71.1** |
| No sample selection (skip step 3) | 69.1 | **94.4** | 93.1 | 73.0 | 76.4 | 86.2 | **33.7** | 68.4 | 54.0 | 57.9 | 71.8 | 70.7 |
| Random sample selection | 69.8 | 94.3 | 93.3 | 72.8 | 76.7 | **86.6** | 33.4 | 68.6 | **54.3** | 57.7 | **73.0** | 70.9 |
| Our sample selection | **69.9** | 94.1 | **93.4** | **73.7** | **76.8** | **86.6** | **33.7** | **68.8** | 54.1 | **58.5** | 72.8 | **71.1** |

Table 6: Additional ablation experiments comparing different training losses (top) and different samples selection strategies (bottom). These experiments use waffleCLIP augmentation instead of the adaptive augmentation.

dataset. For example, scaling $k_1$ by 2 scales the number of training samples by the same amount. The main take-away from Figure 15 is that increasing the number of samples in the training data improves the target accuracy, but only up to a point. The target accuracy saturates at some point, and it is not beneficial to increase $n_{\text{neighbors}}$, $m$, or $k_1$ further.

Tables 4 and 5 perform ablation experiments that justifies paired k-means, adaptive label augmentation, and the diversity preserving loss. we place a check mark next to components being used in the corresponding row. The baseline for paired k-means is k-means clustering of image features only. The baseline for adaptive label augmentation is waffleCLIP. The baseline for the diversity preserving loss ($\lambda = 0.2$) is vanilla cross entropy.

Table 6 performs ablation experiments that compare our loss function against existing loss functions. In this table, soft pseudo labels refer to using the logits of the teacher prediction as the label. We tuned the teacher's temperature parameter. Contrastive loss refers to finetuning with $\mathcal{L}_{\text{CE}} + \mathcal{L}_{\text{contrastive}}$, where the first loss is the cross entropy loss with hard labels, and the second loss is the supervised contrastive loss (Khosla et al., 2020). $\mathcal{L}_{\text{contrastive}}$ is calculated from the image encoder outputs. Training a model using both CE and a contrastive loss in this manner is commonly used in domain generalization, e.g. (Yao et al., 2022). Table 6 also performs ablation experiments justifying our sample selection method in step 3 of Algorithm 2. Our clustering-based sample selection achieves better results than random selection or skipping sample selection.

**Additional Notes** We do not verify the check-sums of the downloaded images, instead we filter out retrieved images where the cosine similarity between the image embedding and query text embedding is very low ($<0.25$). The size of the retrieved datasets is listed in Table 10, and we emphasize again that our framework achieves impressive improvements in accuracy with a small number of retrieved image samples ($<100$K).

**Hardware and Computational Cost**   We ran experiments on a hybrid computing cluster with A40, A100 and L40S GPUs. All experiments require only one GPU at a time. ViT-B/16 experiments require a GPU with 40 GB of memory; ViT-B/14 experiments require a GPU with 80 GB of memory. The paired k-means algorithm was run once on a 20M subset of LAION. This took one hour. Adding all of LAION-2B to the index takes approximately one day. The 4-bit indices require approximately 850 GB of disk space. For Algorithm 2, using ImageNet-1K as an example, we augment each of the 1000 class names 16 times, for a total of 16,000 queries. The retrieval step took 30 seconds in total. For each query, we retrieve the 64 nearest neighbors, but this is not any slower than retrieving only one nearest neighbor when using FAISS Douze et al. (2024) for approximate nearest neighbors search. Upon retrieval, the 64 nearest neighbors are already ranked by similarity to the query. The implementation of step 2 only compares the rank of each image relative to the queries that retrieved it. This finished in 55 seconds for the ImageNet-1K target task. Clustering (step 3) then took 9 minutes and 20 seconds on one CPU, but could be easily sped up using a GPU implementation of k-means. Finally, downloading the 96,000 selected images took 158 seconds.

| | | | DomainNet | | | | | | | Terra Incognita | | |
|---|---|---|---|---|---|---|---|---|---|---|---|---|
| | C | I | P | Q | R | S | **Mean** | 100 | 38 | 43 | 46 | **Mean** |
| **Open-AI CLIP ViT-B/16** | | | | | | | | | | | | |
| CLIP ZS | 71.4 | 47.1 | 66.2 | 13.8 | 83.4 | 63.4 | 57.6 | 51.5 | 26.1 | 34.1 | 29.3 | 35.2 |
| waffleCLIP | 73.0 | 52.0 | 68.3 | 14.0 | 84.9 | 65.8 | 59.7 | 54.2 | 29.5 | 36.4 | 30.6 | 37.7 |
| Random Descriptors | 73.5 | 51.0 | 67.6 | 14.6 | 84.7 | 65.9 | 59.6 | 51.3 | 21.7 | **36.7** | 28.8 | 34.6 |
| Handcrafted Ensemble | 73.7 | 51.2 | 69.3 | 16.0 | 85.0 | 66.2 | 60.2 | 55.4 | 28.5 | 33.4 | **31.0** | 37.1 |
| PromptStyler † | 73.1 | 50.9 | 68.2 | 13.3 | 85.4 | 65.3 | 59.4 | - | - | - | - | - |
| **MUDG (ours)** | **75.3** | **53.8** | **69.8** | **16.4** | **85.6** | **66.6** | **61.2** | **57.7** | **34.6** | 35.7 | 26.8 | **38.7** |
| **Open-AI CLIP ViT-L/14** | | | | | | | | | | | | |
| CLIP ZS | 79.5 | 52.2 | 70.9 | 22.5 | 86.8 | 71.5 | 63.9 | 46.3 | 50.9 | 43.0 | 32.4 | 43.1 |
| waffleCLIP | 80.4 | 56.5 | 72.8 | 22.0 | 88.1 | 73.0 | 65.4 | 45.6 | 45.2 | 43.7 | 31.4 | 41.4 |
| Random Descriptors | 80.6 | 56.0 | 73.4 | 23.3 | 87.9 | 73.2 | 65.7 | 40.9 | 36.3 | 38.5 | 26.3 | 35.5 |
| Handcrafted Ensemble | 81.1 | 55.8 | 73.9 | 24.2 | 87.9 | 73.7 | 66.1 | 47.5 | 50.9 | 41.8 | 30.5 | 42.7 |
| PromptStyler † | 80.7 | 55.6 | 73.8 | 21.7 | 88.2 | 73.2 | 65.5 | - | - | - | - | - |
| **MUDG (ours)** | **81.6** | **58.3** | **74.9** | **24.5** | **88.5** | **74.1** | **67.0** | **53.4** | **53.9** | **46.1** | **32.7** | **46.5** |

Table 7: Terra Incognita and DomainNet results.

| | | PACS | | | | | VLCS | | | |
|---|---|---|---|---|---|---|---|---|---|---|
| | A | C | P | S | **Mean** | Caltech | Labelme | SUN | VOC | **Mean** |
| **Open-AI CLIP ViT-B/16** | | | | | | | | | | |
| CLIP ZS | 97.1 | 99.0 | 99.9 | 88.0 | 96.0 | 99.9 | 68.3 | 75.3 | 85.5 | 82.2 |
| waffleCLIP | 97.3 | 99.0 | 99.9 | 90.3 | 96.6 | 99.9 | 68.6 | 74.4 | 86.3 | 82.3 |
| Random Descriptors | 97.1 | **99.2** | 99.9 | 89.2 | 96.4 | 99.9 | 70.3 | 77.9 | **87.0** | **83.8** |
| Ensemble | 97.6 | **99.2** | 99.9 | 89.9 | 96.7 | 99.9 | **69.1** | 76.4 | 84.2 | 82.4 |
| PromptStyler † | 97.6 | 99.1 | 99.9 | **92.3** | **97.2** | 99.9 | 71.5 | 73.9 | 86.3 | 82.9 |
| **MUDG (ours)** | **97.9** | **99.2** | 99.9 | 90.7 | 96.9 | 99.9 | 65.5 | **78.5** | 86.3 | 82.6 |
| **Open-AI CLIP ViT-L/14** | | | | | | | | | | |
| CLIP ZS | 98.8 | 99.6 | 99.9 | 95.6 | 98.5 | 99.9 | 70.7 | 73.8 | 85.7 | 82.5 |
| waffleCLIP | **99.1** | 99.7 | 100.0 | 95.7 | 98.6 | 99.9 | 70.8 | 74.1 | **87.1** | **83.0** |
| Random Descriptors | 98.9 | 99.6 | 100.0 | 95.6 | 98.5 | 99.9 | 67.6 | **78.0** | 86.4 | **83.0** |
| Ensemble | 98.8 | 99.6 | 100.0 | 95.7 | 98.5 | 99.9 | 65.5 | 76.1 | 85.1 | 81.7 |
| PromptStyler † | **99.1** | 99.7 | 100.0 | 95.5 | 98.6 | 99.9 | **71.1** | 71.8 | 86.8 | 82.4 |
| **MUDG (ours)** | 98.8 | 99.6 | 100.0 | 95.8 | 98.6 | 99.9 | 70.0 | 75.5 | 86.0 | 82.9 |

Table 8: Comparison of our MUDG method with ZS baselines and PromptStyler on PACS and VLCS. Average of three trials. Dataset construction and model training is performed once and evaluated on all domains. † denotes author reported numbers; all other results are our reproductions.

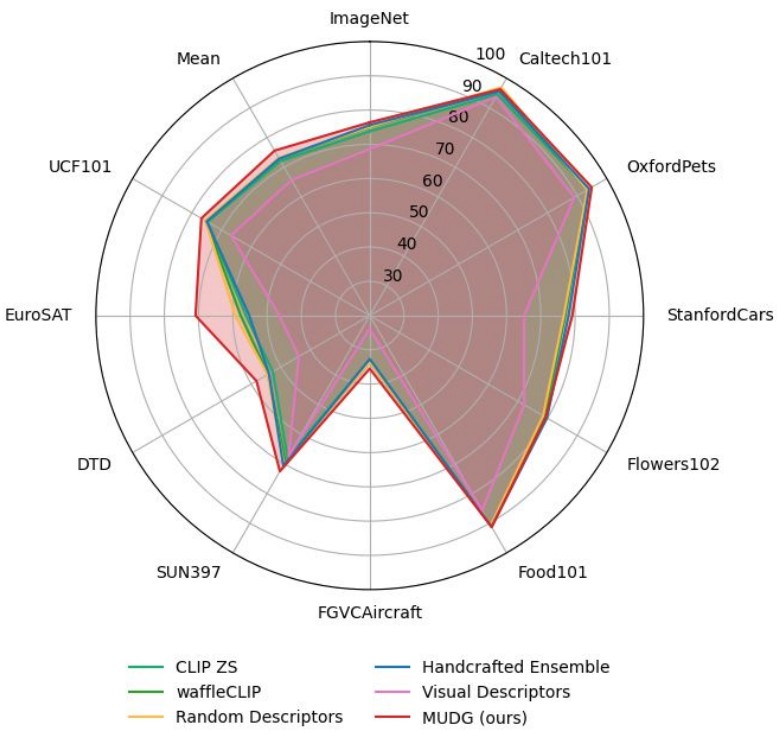

Figure 14: Radial plot of comparisons of the baselines with the pretrained ViT L/14 weights.

## C    DETAILED DESCRIPTION AND MOTIVATION OF ALGORITHM 2

Algorithm 2 consists of a three-stage pipeline presented in Figure 4 (top) used to build a pseudo-labeled subset of the source data.

**Assumptions and notation.**    We are given an unlabeled source dataset $\mathcal{X}_s$ (e.g. LAION-2B English, with text labels discarded). $\mathcal{X}_s$ must be indexed in a joint image-text embedding space by a pair of CLIP encoders $f_{\text{index,text}}$ and $f_{\text{index,image}}$. Both are frozen. We are also given label tokens for the target classification task, formatted as "a photo of a ⟨class name⟩", and denoted as $\{\mathbf{t}_1, ..., \mathbf{t}_c\}$ where $c$ is the number of classes. The goal is to optimize a "student" CLIP model $f_{\text{student}}$ to classify images from the given classes. Note that $f_{\text{student}}$ and $f_{\text{index}}$ can be the same or different models, and we experiment with both possibilities.

### C.1    STEP 1: DIVERSIFIED RETRIEVAL

*Goal:* Retrieve a diverse set of image data for training.

The simplest way to build a dataset from the list of class names is to calculate the text feature for each class and retrieve the nearest neighbors from $\mathcal{X}_s$. This is straightforward, but the results are not promising as shown in the left of Figure 16. The retrieved images are not identical, but contain very little variation. For instance, images of wallets only contain one possible orientation; images of couches only contain stock photos of a perfect couch. Figure 19 (the line with blue x) shows that when trained on these images, the model severely overfits to the retrieved dataset. To diversify the dataset, we augment the query text tokens using the adaptive label augmentation scheme in Section 3.2 of the main paper. From inspecting Figure 16 right, our augmentation seems to capture a broad range of visual variation within each class. We also demonstrate this diversity using t-SNE plots in Figure 17.

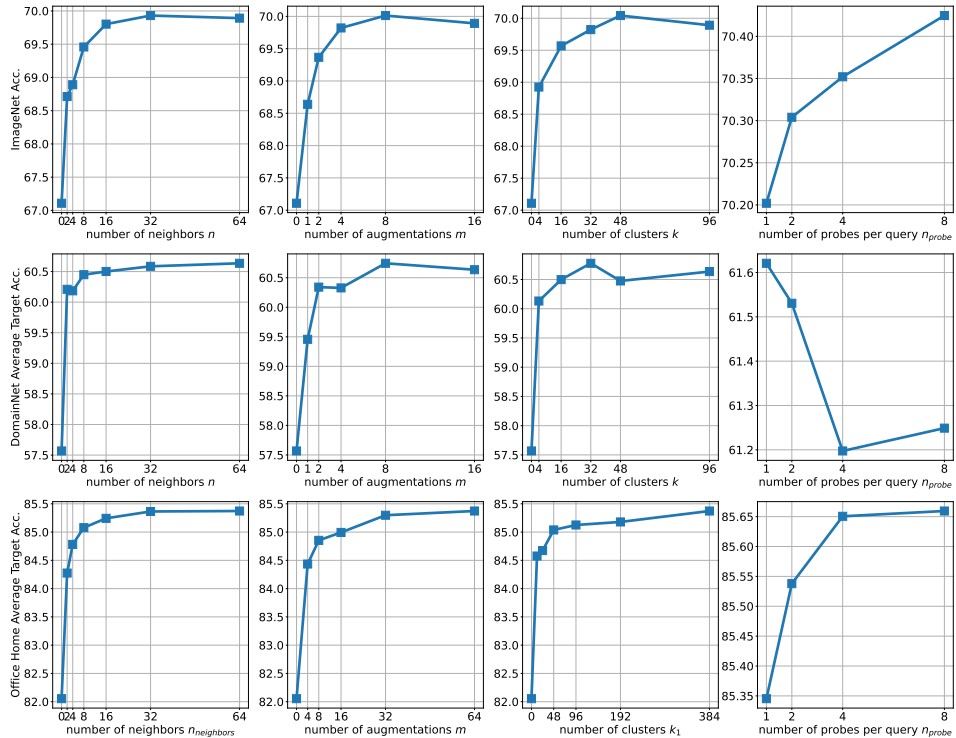

Figure 15: Ablation experiments for varying values of $n_{\text{neighbors}}$, $m$, $k_1$, and $n_{\text{probe}}$. Reference Algorithm 2 in the main paper and Table 10 in the Appendix for default values. Top row: ImageNet; middle row: DomainNet; bottom row: Office Home. Increasing either $n_{\text{neighbors}}$, $m$ or $k_1$ improves the target accuracy by retrieving a larger training set, but these plots show that the accuracy saturates at a certain value. Generally, increasing $n_{\text{probe}}$ also improves the target accuracy.

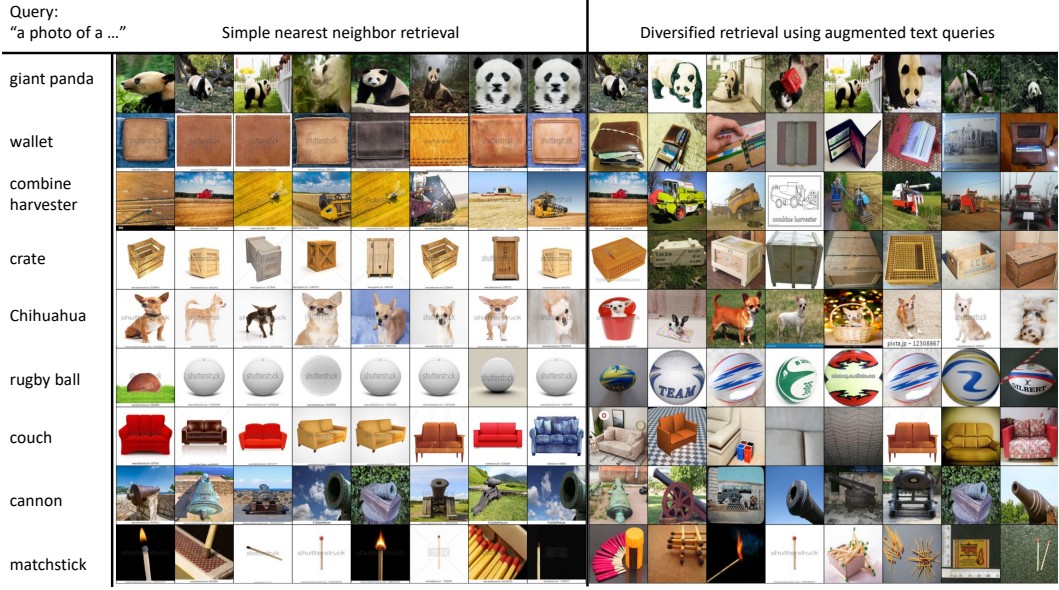

Figure 16: Qualitative results for step 1: diversified retrieval. Left: nearest neighbors to text query in LAION-2B. Right: images retrieved using diversified text features. Images retrieved using diverse queries cover a broader spectrum of appearances in the wild.

| Paired k-means Parameters | |
|---|---|
| $n$ | number of samples in LAION-2B-en |
| $k$ | 131072 |
| number of iterations | 10 |
| **Adaptive Label Augmentation Parameters** | |
| $M$ | 4227 |
| $\{\mathcal{A}_1, ..., \mathcal{A}_M\}$ | unordered ImageNet descriptors from Menon and Vondrick (2022) |
| $k_2$ | 16 |
| $m$ | dataset dependent |

| Finetuning Parameters | | |
|---|---|---|
| | ViT-B/16 | ViT-L/14 |
| Finetune last 3 layers of text and vision encoders | | |
| batch size | 128 | 64 |
| learning rate | 0.00064 | 0.00016 |
| weight decay | 1e-5 | |
| number of iterations ($N$) | dataset dependent | |
| learning rate decay | none | |
| softmax temperature | 25 | |
| optimizer | SGD momentum=0.9 | |
| label smoothing | 0 | |
| EMA weight averaging $\beta$ | 0.995 | |
| text prompt length | 3 | |
| text prompt initialization | "a photo of" | |
| text prompt learning rate multiplier | 10 $\times$ | |
| $\lambda$ | 0.2 | |

| Parameters for Baselines | |
|---|---|
| WaffleCLIP ensemble size | 8 |

Table 9: Training hyperparameters.

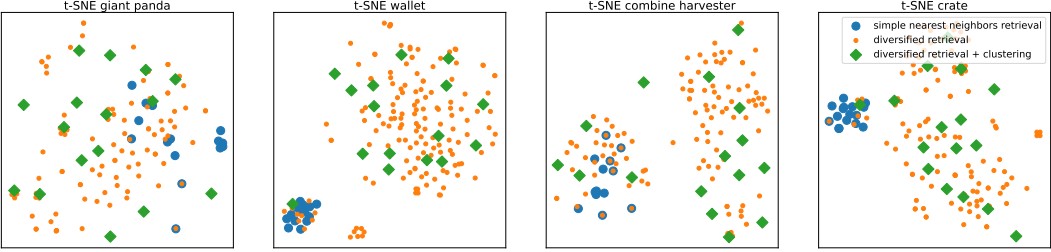

Figure 17: t-SNE plots of image features in the indexing model's embedding space, showing the benefits of steps 1 and 3. Simple nearest neighbor retrieval (blue circle) covers only a small portion of the image distribution for each label. Diversified retrieval (orange dot) covers a broader portion of the image distribution, but contains semantically-redundant samples. After the clustering step (green diamond), the selected image samples are evenly spaced across the entire distribution, and thus the best representation for each label.

## C.2   STEP 2: RANK PSEUDO-LABELING

*Goal:* Mitigate hubness effect.

If each image sample is only retrieved by queries from one label, then pseudo-labeling is trivial. However, there is a large amount of overlap between retrievals from different labels, especially for datasets with a large number of classes or fine-grained concepts. For each image that is retrieved

| Dataset | $n_{\text{neighbors}}$ | $m$ | $k_1$ | $N$ | Actual training dataset size | Query template |
|---|---|---|---|---|---|---|
| ImageNet | 64 | 16 | 96 | 300 | 96K | `a photo of a {}.` |
| Caltech | 64 | 64 | 384 | 100 | 38K | `a photo of a {}.` |
| Pets | 64 | 64 | 384 | 200 | 12K | `a photo of a {} , a type of pet.` |
| Cars | 64 | 16 | 96 | 1000 | 18K | `a photo of a {}.` |
| Flowers | 64 | 64 | 384 | 200 | 31K | `a photo of a {} , a type of flower.` |
| Food | 64 | 64 | 384 | 100 | 34K | `a photo of a {} , a type of food.` |
| Aircraft | 64 | 64 | 384 | 1000 | 26K | `a photo of a {} , a type of aircraft.` |
| SUN | 64 | 16 | 96 | 300 | 38K | `a photo of a {}.` |
| DTD | 64 | 64 | 384 | 200 | 18K | `a photo of a {} texture.` |
| EuroSAT | 64 | 64 | 384 | 200 | 3K | `a photo of a {} , from a satellite.` |
| UCF | 64 | 64 | 384 | 200 | 37K | `a photo of a person doing {}.` |
| ImageNet-V2 | 64 | 16 | 96 | 200 | 96K | `a photo of a {}.` |
| ImageNet-Sketch | 64 | 16 | 96 | 200 | 96K | `a photo of a {}.` |
| ImageNet-A | 64 | 16 | 96 | 200 | 19K | `a photo of a {}.` |
| ImageNet-R | 64 | 16 | 96 | 200 | 19K | `a photo of a {}.` |
| DomainNet | 64 | 16 | 96 | 200 | 33K | `a photo of a {}.` |
| Office Home | 64 | 64 | 384 | 200 | 25K | `a photo of a {}.` |
| PACS | 64 | 64 | 384 | 100 | 3K | `a photo of a {}.` |
| VLCS | 64 | 64 | 384 | 50 | 2K | `a photo of a {}.` |
| Terra Incognita | 64 | 64 | 384 | 100 | 3K | `a photo of a {} , from a camera trap.` |

Table 10: Dataset-specific hyperparameters, reference Algorithm 2 in the main paper. $n_{\text{neighbors}}$ is number of nearest neighbors to be retrieved; $m$ is number of text augmentations; $k_1$ is number of k-means clusters; $N$ is number of training iterations.

| Augmentation $\mathcal{A}$ | Loss (Eq. 6) |
|---|---|
| `a photo of a {}, which may have multiple settings (low, medium, high).` | 0 |
| `a photo of a {}, which often has a design or logo.` | 0 |
| `a photo of a {}, which has people often in close proximity.` | 0 |
| `a photo of a {}, which is a gradually increasing or decreasing diameter.` | 0 |
| `a photo of a {}, which has usually rectangular or square in shape.` | 0 |
| ... | ... |
| `a photo of a {}, which is a piece of clothing.` | 16 |
| `a photo of a {}, which is a piece of armor.` | 16 |
| `a photo of a {}, which is a pie dish.` | 16 |
| `a photo of a {}, which is a phone receiver with a cord.` | 16 |
| `a photo of a {}, which is a pen with a decorative band or ring.` | 16 |

Table 11: Qualitative results for our adaptive text augmentation on ImageNet. Losses are calculated based on Equation 6. $k_2 = 16$. The loss value is an integer in range $[0, k_2]$.

by multiple queries, we can assign it either (1) the label of the closest text feature as measured by their cosine similarity, or (2) the label of the text feature to which it is ranked the highest. We choose the latter option (detailed concretely in Algorithm 2) to address the well-known hubness effect (Radovanovic et al., 2010). In simple terms, hubs are samples in the dataset which tend to be closer to other samples in a high-dimensional embedding space, regardless of relevance. Specific to our application, a "hub" text feature is one that is close to a disproportionately large number of image samples, resulting in a large number of image samples being assigned the hub label. In other words,

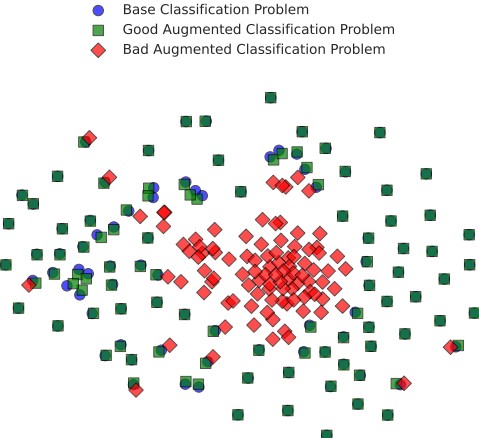

Figure 18: t-SNE visualization corresponding to Figure 3 in the introduction. We randomly select 100 ImageNet class names and compute the text features corresponding to the "a photo of a {}" prompt (referred to as the base classification problem). Additionally, we use a bad augmentation selected from Table 11 with a loss of 16, and a good augmentation with a loss of 0. The features are aggregated and visualized in this figure. The augmentation with higher loss does not preserve the variance of the classifier prototypes, likely reducing the ZS accuracy and diminishing its effectiveness.

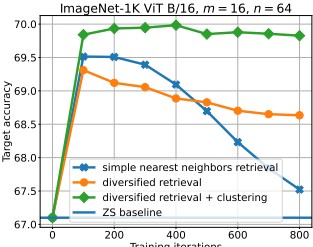

Figure 19: Target accuracy vs. training iterations for the datasets corresponding to Figure 17 (colors match). This confirms our intuition that both the diversified retrieval and clustering steps are necessary. $n$ here refers to $n_{\text{neighbors}}$.

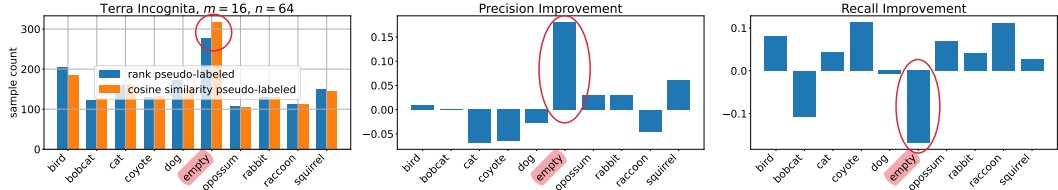

Figure 20: Hubness effect and the value of pseudo-labeling based on rank (step 2). The x-axis labels are the label names for Terra Incognita. The label "empty" is a hub because about 50 more images were labeled as empty when cosine similarity is used instead of rank (left bar plot). The right two bar plots show the precision and recall improvement *of rank labeling over cosine similarity labeling*, after clustering and training. Rank labeling improves precision for images labeled as empty while improving the recall for most animal images. *This is desireable*: The cost of mislabeling an animal image as empty is much greater than the cost of mislabeling an empty image. $n$ here refers to $n_{\text{neighbors}}$.

the pseudo-label is biased towards any hubs in the label space when cosine similarity is used directly. However, when we use rank to assign labels, the hub label cannot be overused because closeness to the hub is determined by rank relative to other image samples.

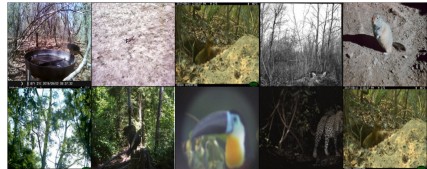

Figure 21: A selection of images from the 50 that were labeled as "empty" by cosine similarity but as one of the animals by rank.

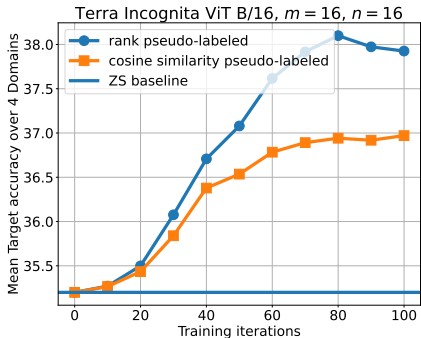

Figure 22: Target accuracy vs. training iterations for the two datasets constructed for Figure 20 left. Colors match. This confirms our intuition that rank labeling improves overall target accuracy in addition to the good precision-recall properties in Figure 20 right. $n$ here refers to $n_{\text{neighbors}}$.

Not all datasets have hubs, but we found that the Terra Incognita dataset illustrates the effect perfectly. This dataset contains camera trap images of different animals, and the labels are the animal names along with "empty" for empty images. As a case study, we retrieve images from LAION-2B using step 1 and the query: "a photo of a ⟨class name⟩, from a camera trap.". We then compare pseudo-labeling using cosine similarity versus using rank. The left bar plot in Figure 20 shows that cosine similarity pseudo-labeling assigns some images the "empty" label, which are labeled as one of the animals when using rank. Figure 21 displays examples of these images. For this dataset, "empty" likely functions as a hub, since many camera-trap images are mostly empty, especially if the animal is small. We verify in the two right bar plots of Figure 20 that using rank pseudo-labeling improves the recall of most animal images at the expense of decreasing the recall of empty images. This is a favorable trade-off for this application. We further verify in Figure 22 that rank pseudo-labeling improves the overall accuracy as well, compared to cosine similarity labeling.

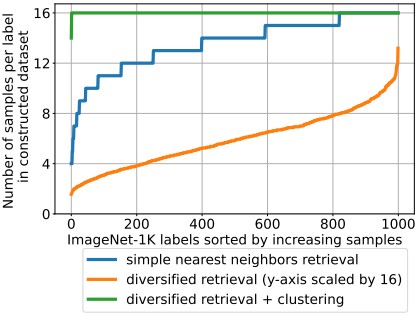

Figure 23: Label distribution of datasets constructed before and after clustering (step 3). For this figure, $m = n_{\text{neighbors}} = k_1 = 16$.

## C.3 STEP 3: CLUSTERING

*Goal:* Select representative samples and balance the label distribution.

Referring back to Figure 19, note that the dataset resulting from diversified retrieval (in orange) actually lowers target accuracy on ImageNet when used to train the student CLIP model, despite containing a large number of samples ($\mathcal{O}(mcn_{\text{neighbors}})$). This stems from two problems: (1) Some images are semantic-duplicates as evident by the small clusters of orange dots in Figure 17, e.g. pictures of the same object in different orientations. (2) The dataset is imbalanced as shown by the orange distribution over labels in Figure 23. This is simply caused by asymmetries in the retrieval and download process (e.g. dead links, linked image changed since dataset creation, etc.). As a result, the training process overfits to dominant semantic-duplicate images and the pseudo-label distribution; both are artifacts of the dataset construction process.

To address both of the above issues, we first use k-means clustering to cluster the image features in the embedding space of the indexing model into $k_1 << mcn_{\text{neighbors}}$ clusters, then randomly select an image from each cluster. If $k_1$ is chosen conservatively, semantic duplicates fall into a single cluster, and only one can be selected for the final training set. Additionally, each label should have $k_1$ training samples. Figure 23 illustrates the final balanced label distribution in green, and Figure 19 shows the corresponding target accuracy improvements in matching colors. For reference, ImageNet-1K has $c = 1000$ labels, and we found $m = 16$, $n_{\text{neighbors}} = 64$ and $k_1 = 48$ to yield good results.

