# OpenReview forum: "Multimodal Unsupervised Domain Generalization by Retrieving Across the Modality Gap"
_ICLR.cc/2025/Conference — ICLR 2025 Poster_

### Official Review · Reviewer_6mKA · 2024-10-30

**Soundness:** 4
**Presentation:** 3
**Contribution:** 4
**Rating:** 8
**Confidence:** 2

**Summary:**

This paper studies multimodal unsupervised domain generalization (MUDG), where only task-agnostic source data and target label names are required. Accurate yet diverse retrieving image samples that match the specific text description for a certain class name becomes the crux. Towards the former, the authors propose paired k-means, maximizing the probability of a text query and its closest image sample belonging to the same Voronoi cell by updating the centroids to be in the query distribution. As for the latter, the authors develop a heuristic augmentation strategy. Experiments under various evaluation protocols demonstrate the effectiveness of the proposed method.

**Strengths:**

- This paper is well-written overall, and the motivation is quite reasonable.
- Each proposed component is supported by sufficient theoretical and empirical analysis.
- Experiments are sufficient.

**Weaknesses:**

- Table 1 should be placed before Figure 1 as Table 1 was mentioned before Figure 1.
- The bottom right subfigure of Figure 1 was an illustration. Could you provide a quantitative comparison between good augmentation and bad ones (such as using t-SNE visualization)?
- Ablations in Tables 4 and 5 can not reflect the effectiveness of each component, as the improvements are marginal. Could you report the results of statistical testing to see if the improvements are significant or not?

**Questions:**

I have no further questions. Please refer to the "Weaknesses" section.

---

> ### Author Response · Authors · 2024-11-23
> **Author Response**
>
> We thank you for your positive and encouraging review of our submission! Please find our responses to the weaknesses below:
>
> *Response to weakness 1:* We repositioned the figures as requested.
>
> *Response to weakness 2:* Thank you for the suggestion! We added Figure 18 to page 27 of the revised manuscript. This is a t-SNE visualization corresponding to Figure 3 in the introduction. We randomly select 100 ImageNet class names and compute the text features corresponding to the "a photo of a {}" prompt (referred to as the base classification problem). Additionally, we use a bad augmentation selected from Table 11 with a loss of 16, and a good augmentation with a loss of 0. The features are aggregated and visualized in this figure. The augmentation with higher loss does not preserve the variance of the classifier prototypes, likely reducing the ZS accuracy and diminishing its effectiveness.
>
> *Response to weakness 3:* We appreciate your suggestion to include statistical testing for the significance of the ablation results. Unfortunately, due to resource limitations, we were unable to perform enough runs of each ablation experiment to conduct meaningful statistical tests. In Tables 4 and 5 of the submission, we reported the average of three random trials. We did not include the standard deviation, as the number of trials was insufficient to provide meaningful error bars.

---

> > ### Comment · Reviewer_6mKA · 2024-11-27
> > **Post Rebuttal Comments**
> >
> > I appreciate the rebuttal. I will keep my rating and recommend acceptance.
> >
> > However, statistical testing to see if the improvements are significant or not is strongly encouraged in further revisions, e.g., the camera-ready version.

---

> > > ### Author Response · Authors · 2024-11-27
> > >
> > > Thank you for acknowledging our rebuttal!

---

### Official Review · Reviewer_gdYC · 2024-10-31

**Soundness:** 2
**Presentation:** 2
**Contribution:** 3
**Rating:** 6
**Confidence:** 4

**Summary:**

This paper presents a novel approach to multimodal unsupervised domain generalization that leverages a large unlabeled source dataset for finetuning, without requiring a direct relationship with the target task. Key contributions include (1) a theoretically derived paired k-means algorithm that enhances nearest neighbor recall, (2) an adaptive text augmentation scheme to improve zero-shot accuracy, and (3) two additional components that boost target accuracy. Experimental results demonstrate consistent accuracy improvements across 20 diverse datasets.

**Strengths:**

This paper is well-motivated and appears to be reproducible based on the author's code. The author discusses the challenges of cross-modal retrieval from a theoretical perspective and proposes methods to address the issue of low recall. Additionally, the author proposes an adaptive text augmentation scheme to tackle unsupervised domain generalization in multimodal contexts. Comprehensive analyses, comparative experiments, and insightful visualizations highlight the appropriateness of the method's genuine effectiveness.

**Weaknesses:**

1. While the methodology appears to be fundamentally solid, my primary concern regarding this paper is the clarity of expression and how it affects overall readability. Several figures in the paper, such as Figure 1, present ambiguous meanings, and others, like Figures 4 and 5, lack sufficient quality. Additionally, there are issues with the details in the descriptions, including an overuse of abbreviations and numerous typos in the introduction. These factors contribute to a challenging reading experience for the audience.
2. I believe a more detailed introduction to the task setting and motivation is necessary. Although the author mentions that MUDG is more practical, it seems to be primarily a combination of text-image representation and UDG. Could the author elaborate on specific application scenarios?
3. In terms of retrieval, I’m curious why only the first-level k-means is utilized instead of employing a fine quantization scheme at the second level. What are the implications of this choice, and might it impact the recall rate?
4. In Table 2, the proposed method does not seem to achieve superior performance across many datasets. Given that MUDG is being addressed so specifically, what could explain the lack of a more significant effect? What might be the underlying reasons for this?
5. Lastly, regarding the key hyper-parameters listed in Table 9, how are these parameters determined? Are they sensitive to variations, and how do they influence the performance of the method? This aspect warrants further discussion.

**Questions:**

See 'weakness' for details. A more thorough introduction to the motivation and clarification of the methods are needed. If the author can improve the presentation of the paper in the next version, I would be happy to raise my score!

---

> ### Author Response · Authors · 2024-11-23
> **Author Response**
>
> Thank you for the very thorough review of our work! We have made numerous revisions and hope that you can re-evaluate based on the revisions and our responses below.
>
> *Response to weakness 1:*
>
> We thoroughly revised the introduction and figures according to your suggestions.
> We **broke Figure 1 into three figures**, each with revised captions. We then repositioned each figure to be closer to the relevant text.
> We **improved the quality of Figures 7 and 8** (formerly Figures 4 and 5).
>
> *Response to weakness 2:*
>
> We **added Figure 1** to create a more intuitive visualization of the task setting.
> We then added a paragraph beginning at line 80 detailing **a specific application scenario.**
>
> To elaborate further on the use case described in the revised introduction, traditional domain generalization methods, including unsupervised ones, rely on training datasets tailored to a specific classification task. In contrast, our MUDG paradigm removes the requirement for training data to be task-specific. This flexibility allows the same source dataset to support multiple related classification problems without the need for manual dataset re-engineering, enabling broader applicability and reducing the overhead associated with task-specific data preparation.
>
> Consider e-commerce applications where products often require classification in multiple directions. For instance, a product like shoes might need to be categorized based on material (e.g., leather vs. synthetic), style (e.g., casual vs. formal), or usage (e.g., indoor vs. outdoor). Traditionally, this would involve manually curating datasets tailored to each specific classification task. Our approach automates this process by enabling a single, task-agnostic dataset to support various classification tasks.
>
> *Response to weakness 3:*
>
> From our understanding of the literature, a two-level data structure is generally used with both coarse and fine quantization components. The coarse quantization decreases the query time by limiting the amount of samples that must be searched through. The main goal of fine quantization is to compress the embeddings, although it does also affect recall and latency. We were primarily interested in studying the coarse quantization component, because it is clearly impacted by the modality gap issue.
>
> In our experiments, we chose not to explore sophisticated fine quantization techniques (such as product quantization) because of the following technical issue. In FAISS, fine quantization operates on the residuals of the coarse quantization (i.e. the difference vector between the sample embedding and the closest centroid). Unfortunately, our algorithm repositions the centroids away from the sample embeddings. This increases the length of the residual vectors and decreases the accuracy of the fine quantization. A simple fix would be to store centroids in both the image space and the query space, and calculate residual vectors with respect to image centroids. We believed that including this would detract too much from the primary focus of our work.
>
> *Response to weakness 4:*
>
> We acknowledge that the proposed method does not demonstrate superior performance across all datasets in Table 2. This outcome can be attributed to several underlying factors. We primarily tuned our method on ImageNet, so the results are better on datasets with similar characteristics, such as SUN and DomainNet. Herding is a challenging coreset selection baseline that is good at selecting samples which best approximate the PDF of the underlying data distribution. Consequently, the herding method might outperform in scenarios where the retrieved dataset is similar in distribution to the target dataset. This is the case for target datasets assembled by querying a search engine, with minimal post-processing. For datasets (such as Office Home, DomainNet, and the different versions of ImageNet in Table 3) where the curators attempted to gather a wide range of target domains, our method is likely to outperform the coreset selection baselines. We emphasize that our method still achieves the best overall target score compared to these challenging baselines.
>
> *Response to weakness 5:*
>
> The finetuning parameters in Table 9 (such as learning rate, which layers to freeze, and EMA constant) were determined by training on a random 16-shot subset of ImageNet and validating on another random 16-shot subset. These parameters are not specific to our problem setting. The other parameters, listed in Table 10, are dataset-dependent and were determined using a validation set.
>
> We performed an analysis of the sensitivity of important hyperparameters in Figure 15. We observed that increasing either $n_{neighbors}$, $m$ or $k_1$ improves the target accuracy by retrieving a larger training set, but the accuracy saturates
> at a certain value. Generally, increasing $n_{probe}$ also improves the target accuracy.

---

> > ### Comment · Reviewer_gdYC · 2024-11-23
> >
> > Thank you for the author's response and revisions. I think the new version presents the statement and motivations much more clearly than before, and I am happy to consider raising the score. However, I still feel that the images used for the e-commerce classification in the motivation do not align with the image datasets used in the experiments. In the future, using data from relevant application scenarios may be more convincing.

---

> > > ### Author Response · Authors · 2024-11-25
> > >
> > > Thank you for reading our rebuttal and raising your score!

---

### Official Review · Reviewer_TM1x · 2024-11-01

**Soundness:** 2
**Presentation:** 3
**Contribution:** 3
**Rating:** 6
**Confidence:** 3

**Summary:**

This paper presents a novel approach to multimodal unsupervised domain generalization (MUDA), where the source data is both unlabeled and task-agnostic. First, it theoretically examines the issues with existing cross-modal approximate nearest neighbor search methods. Next, it proposes a paired k-means algorithm to enhance neighbor search efficiency. Finally, we introduce a text augmentation scheme aimed at improving zero-shot accuracy.

**Strengths:**

1.The paper introduces the MUDG setting and presents a novel paired k-means algorithm to address the cross-modal nearest neighbor problem.

2.The paper provides a thorough theoretical analysis demonstrating the limitations of existing cross-modal nearest neighbor methods, supported by extensive experimental validation

3.Figures are effectively used to illustrate empirical observations, the improvements brought by paired k-means, and the visualization of Voronoi cells, enhancing the understanding of the proposed method.

**Weaknesses:**

1.The paper compares MUDG setting to zero-shot (ZS) and source-free domain generalization (SFDG) setting in Tables 1. ZS and SFDG impose more stringent conditions, while MUDG relies on task-agnostic source data, making it less practical in some contexts. Therefore, comparing MUDG methods with SFDG or ZS methods may not be entirely fair.

2.The paper lacks a comparison of the computational resources and time required by its proposed method in relation to other SFDG and ZS methods. Since this setting involves large-scale datasets during training, it may significantly increase the computational resources and time needed

3.In Table 2, the proposed method shows only a slight increase of 0.6% compared to the second-best method.

4.In the ablation experiments (Table 5), when adaptive augmentation is used alone, there is a drop of about 1%, with some datasets not even reaching that. This indicates that the paired k-means method, which is the main contribution of this work, is not effective.

**Questions:**

In Table 3, there are no MUDG methods included. Could you combine the PromptStyler method with the proposed methods for a fair comparison?

---

> ### Author Response · Authors · 2024-11-23
> **Author Response**
>
> We thank you for your critical and encouraging review of our submission! We are happy that you found our experiments to be expensive and the visualizations to be helpful. Please find our responses to the weaknesses below:
>
> *Response to weakness 1:*
>
> In reference to Tables 2 and 3, we agree with the reviewer that the MUDG setting is less stringent than either the ZS or SFDG setting. However, we do also have many MUDG baselines, and the comparisons with the MUDG baselines are fair. We clearly label which method is under which setting, so as not to misrepresent the results. However, some readers familiar with the literature will expect to see comparisons to zero-shot methods to establish a lower bound. This presentation of results is inline with prior work, including PromptStyler. We also include an upper bound in Table 2.
>
> We emphasize that the goal of including different ZS baselines is to present a well-calibrated lower bound, not to misrepresent results.
>
> *Response to weakness 2:*
>
> We agree. The MUDG setting is more computationally expensive than either SFDG or ZS. This is an expected tradeoff.
>
> As a quantitative example, for ImageNet-1K, we augment each of the 1000 class names 16 times, for a total of 16,000 queries. The retrieval step took 30 seconds in total. For each query, we retrieve the 64 nearest neighbors, but this is not any slower than retrieving only one nearest neighbor when using FAISS [3] for approximate nearest neighbors search. Upon retrieval, the 64 nearest neighbors are already ranked by similarity to the query. The implementation of step 2 only compares the rank of each image relative to the queries that retrieved it. This finished in 55 seconds for the ImageNet-1K target task. Clustering (step 3) then took 9 minutes and 20 seconds on one CPU, but could be easily sped up using a GPU implementation of k-means. Finally, downloading the 96,000 selected images took 158 seconds.
>
> | **ImageNet-1K Target Task** | **Computation Cost**      |
> |-----------------------------|---------------------------|
> | Retrieval (step 1)          | 30 seconds               |
> | Pseudolabeling (step 2)     | 55 seconds               |
> | Clustering (step 3)         | 9 minutes 20 seconds     |
> | Downloading images          | 158 seconds              |
>
> Please note that computational efficiency of the retrieval process was not a goal of our work.
>
> [3] [FAISS](https://github.com/facebookresearch/faiss)
>
> *Response to weakness 4:*
>
> We think that there might be a misunderstanding of the ablation experiment in Table 5. This experiment intends to show that removing any of the three components (paired k-means, adaptive augmentation, or diversity preserving loss) from our method decreases the overall target accuracy. Removing the diversity-preserving loss decreases the average target accuracy by 0.3% and 0.4% on DomainNet and OfficeHome, respectively.  Removing the paired k-means retrieval decreases the average target accuracy by 0.2% and 1.4% on DomainNet and OfficeHome, respectively.  Removing the adaptive augmentation decreases the average target accuracy by 0.3% and 0.2% on DomainNet and OfficeHome, respectively.
>
> *Response to Question:*
>
> The reviewer suggests to include MUDG baselines in Table 3. We are unable to reproduce the PromptStyler results because they did not release code. However, we do agree with the reviewer that MUDG baselines are needed in Table 3. We ran the additional experiments; the revised rows for the ViT-B/16 model size is as follows:
>
> "IN" stands for ImageNet. "OH" stands for Office Home.
>
> | **Method**        | **Setting** | **IN-V2** | **IN-S** | **IN-A**  | **IN-R**  | **IN-Mean** | **OH-A** | **OH-C**  | **OH-P**  | **OH-R**  | **OH-Mean** |
> |--------------------|------|-----------------|------------|--------|--------|----------|------------------|--------|--------|--------|----------|
> | **Margin**                   | MUDG |  59.6            | 46.4       | 45.3   | 75.8   | 56.8     | 83.7             | 72.9   | 88.9   | 88.5   | 83.5     |
> | **Least Confidence**  | MUDG |  58.0            | 45.6       | 44.7   | 75.1   | 55.8     | 84.5             | **74.2**   | 90.0   | 89.7   | 84.6     |
> | **Entropy**                  | MUDG |  59.2            | 46.4       | 44.7   | 75.6   | 56.5     | 84.2             | 73.3   | 89.0   | 89.0   | 83.9     |
> | **Herding**                  | MUDG |  62.4            | 49.3       | 49.0   | 78.8   | 59.9     | 84.2             | 73.8   | 89.2   | 89.2   | 84.1     |
> | **K-Center Greedy**   | MUDG |  62.0            | 48.7       | 49.7   | 78.6   | 59.7     | 84.6             | 74.1   | 90.2   | 89.9   | 84.7     |
> | **MUDG (ours)**         | MUDG |  **63.6**            | **50.4**       | **51.5**   | **80.1**   | **61.4**     | **85.9**             | 73.3   | **92.0**   | **91.4**   | **85.7**     |

---

> > ### Comment · Reviewer_TM1x · 2024-11-25
> >
> > Thank you for the authors' feedback. Most of my concerns have been addressed; however, I still have reservations about the motivation of this setting. Therefore, I am maintaining my original positive score.

---

> > > ### Author Response · Authors · 2024-11-25
> > >
> > > Thank you for reading our rebuttal and the positive feedback!

---

### Official Review · Reviewer_qfP3 · 2024-11-03

**Soundness:** 4
**Presentation:** 3
**Contribution:** 3
**Rating:** 8
**Confidence:** 4

**Summary:**

This paper introduces Multimodal Unsupervised Domain Generalization (MUDG), a novel framework that leverages large-scale unlabeled datasets (like LAION-2B) to improve model generalization on target domains. The paper makes three main contributions: (1) proposing paired k-means algorithm to improve cross-modal retrieval accuracy by updating centroids in query space rather than image space; (2) designing an unsupervised text augmentation scheme that adaptively selects appropriate descriptors for target labels; and (3) introducing a clustering-based sample selection strategy and diversity-preserving loss function to construct more representative training datasets. The method demonstrates significant improvements over existing zero-shot learning and source-free domain generalization baselines across 20 diverse datasets.

**Strengths:**

1. The paper proposes a novel MUDG framework that effectively leverages large-scale unlabeled datasets for domain generalization through cross-modal retrieval and adaptive text augmentation.
2. The paired k-means algorithm effectively addresses the modality gap in cross-modal retrieval, supported by rigorous theoretical analysis and empirical validation.
3. Extensive experiments across 20 diverse datasets demonstrate significant improvements over state-of-the-art baselines in zero-shot learning and domain generalization, showing strong practical value.

**Weaknesses:**

1. While the paper introduces MUDG as a new paradigm beyond UDG, it would benefit from a visual comparison/illustration showing the differences between DG, UDG, and MUDG. Moreover, the necessity and practical benefits of this new task definition are not sufficiently justified - authors should more explicitly demonstrate why existing frameworks like UDG are inadequate and how MUDG addresses real-world challenges.
2. In this paper, authors explore a way to retrieval task-agonistic source dataset to assist in model learning in the target domain. What is the intuition behind this operation? Is the assumption reasonable?
3. The writing of the Introduction Section is confusing: authors present a group of operations and six contributions in the section of introduction. However, what is the core challenge and the corresponding solution method proposed in this paper?

**Questions:**

For detailed questions and suggestions, please refer to the Weaknesses section

---

> ### Author Response · Authors · 2024-11-22
> **Author Response**
>
> Thank you for the great suggestions! We appreciate your highlighting of our submission's "rigorous theoretical analysis and empirical validation" and "strong practical value". Here are our responses to the weaknesses:
>
> *Response to weakness 1:*
>
> We have **added Figure 1** comparing DG, UDG, and MUDG as suggested.
> Additionally, we included **a paragraph beginning at line 80** detailing a specific application scenario to illustrate the practical relevance of MUDG.
>
> Traditional domain generalization methods, including unsupervised ones, rely on training datasets tailored to a specific classification task. In contrast, our MUDG paradigm removes the requirement for training data to be task-specific. This flexibility allows the same source dataset to support multiple related classification problems without the need for manual dataset re-engineering, enabling broader applicability and reducing the overhead associated with task-specific data preparation.
>
> To elaborate further on the use case described in the revised introduction, consider e-commerce applications where products often require classification in multiple directions. For instance, a product like shoes might need to be categorized based on material (e.g., leather vs. synthetic), style (e.g., casual vs. formal), or usage (e.g., indoor vs. outdoor). Traditionally, this would involve manually curating datasets tailored to each specific classification task. Our approach automates this process by enabling a single, task-agnostic dataset to support various classification tasks.
>
> *Response to weakness 2:*
>
> We would like to reiterate that our approach is only applicable to scenarios where the target visual concepts are present in the source dataset. This assumption is reasonable across all the benchmarks explored in our experiments. The LAION dataset, which we used as our source, represents a broad subset of images available on the web. For traditional computer vision benchmarks like ImageNet, retrieving relevant images from this dataset poses no challenge.
>
> For more fine-grained target datasets, such as airplane classification, relevant images still exist within the source dataset. However, the suboptimal alignment between the labels and images in the embedding space could reduce the effectiveness of our framework.
>
> In the e-commerce use case described earlier, the validity of our assumptions is clear.  A company like Amazon, for example, likely has images for nearly every product imaginable. While these images may not be labeled for a specific classification task, their availability in a task-agnostic dataset supports our framework's applicability.
>
> **Intuition:** The intuition behind retrieving relevant images and fine-tuning the model on pseudo-labeled data aligns closely with the principles of semi-supervised learning. Even though the constructed dataset lacks ground-truth labels, the retrieved images represent high-confidence samples for each class. Training on these samples enables the model to achieve high accuracy, with the caveat of inheriting biases present in the model used for retrieval.
>
> *Response to weakness 3:*
>
> We agree that the list of six contributions can be confusing to the reader. In our revision, we reduced this to two bullet points. **The core challenge of our work is how to effectively retrieve images from the source dataset for the downstream task.** To address this challenge, we presented a method for more accurate cross-modal search (the first contribution) and a procedure for diversified retrieval (the second contribution). The corresponding paragraphs in the introduction are clearly labeled. We hope that this improves the organization and readability of the introduction.

---

> ### Comment · Reviewer_qfP3 · 2024-11-30
>
> Thank you for the revisions and for taking the time to address my concerns. The added comparison and clearer explanation helped me better understand the importance of MUDG. I’ve decided to raise my score to 8. Appreciate your hard work!

---

### Author Response · Authors · 2024-11-22
**Rebuttal Revision**

We sincerely thank the reviewers for your helpful feedback.

Based on the initial reviews, the main weakness of our submission was the presentation and readability of the introduction. In particular reviewer qfP3 and gdYC wanted to see more explanation of the multimodal unsupervised domain generalization (MUDG) setting. We tried our best to address these shortcomings in the current revision. The revisions are highlighted in blue. We made the following changes:

- Added Figure 1 to provide a visual comparison between our problem setting and prior domain generalization settings. (Reviewers qfP3
and gdYC)
- Figures 2, 3, and 4 used to be one figure. We broke this figure up in this revision and repositioned them to be adjacent to the relevant text. We also added more descriptive captions. (Reviewer gdYC)
- We added a paragraph (line 80) to the introduction describing an example real-world application of our work. (Reviewers qfP3
and gdYC)
- We condensed our contributions to two concise bullet points. (Reviewer qfP3)
- We improved the quality of figures 7 and 8. (Reviewer gdYC)

We will also respond to the reviews individually. Please let us know if the manuscript can be improved further.

---

### Meta-Review · Area_Chair_7ynQ · 2024-12-15

**Metareview:**

This paper addresses a unique challenge of using source data from various domains that were not specifically created for the target task. The proposed approach builds on self-taught learning concepts and incorporates multiple modalities in a distinct way. Experiments on 20 diverse datasets show consistent accuracy improvements over advanced methods such as name-only transfer, source-free domain generalization (DG), and zero-shot (ZS) approaches. During the rebuttal process, the authors added further experimental results, clarified the problem setup, and refined the core ideas. These updates received positive feedback, with reviewers expressing strong approval and satisfaction with the revisions.

**Additional Comments On Reviewer Discussion:**

During the rebuttal process, all concerns are well addressed, for example, the authors added further experimental results, clarified the problem setup, and refined the core ideas.

---

### Decision · Program_Chairs · 2025-01-22

Accept (Poster)